# QiMeng-PerceptOS: Semantic-Aware Kernel Optimization for OS-Intensive Workloads via Hardware-Software Alignment

**Huilai Chen** [1 2] **Yuanbo Wen** [1 2] **Liangfeng Li** [2 3] **Shaohui Peng** [2 4] **Jingzhe Zhu** [2 4] **Jun Bi** [1] **Xuzhi Zhang** [2 4] **Qi Guo** [1 2] **Ling Li** [2 4] **Yunji Chen** [1 2]

## Abstract

Optimizing OS kernels for specific applications is vital for peak performance, yet existing LLM-based methods struggle with a semantic mismatch between generalized reasoning and low-level system behaviors. As a result, these static, open-loop approaches suffer from runtime blindness, configuration fragmentation, and search drift, ultimately failing to unlock the system's performance potential. To address this, we propose QiMeng-PerceptOS, an autonomous framework that shifts the paradigm to perception-driven tuning. QiMeng-PerceptOS integrates: (1) a Perception Module that aligns raw telemetry into high-fidelity semantic fingerprints; (2) a Global Search Module utilizing a Bi-level Hierarchical Induction Tree (BHIT) for global navigation and efficient pruning; and (3) a Posterior Enhancement Module to suppress hallucinations via trajectory synthesis. Experiments across diverse workloads show that it achieves significant performance breakthroughs by optimizing kernel configurations, reaching 296.6% of default Redis throughput and surpassing SOTA baselines by 32.6% within only 15 iterations. By establishing a perception-driven closed-loop, QiMeng-PerceptOS provides new insights for fully automated, large-scale system optimization.

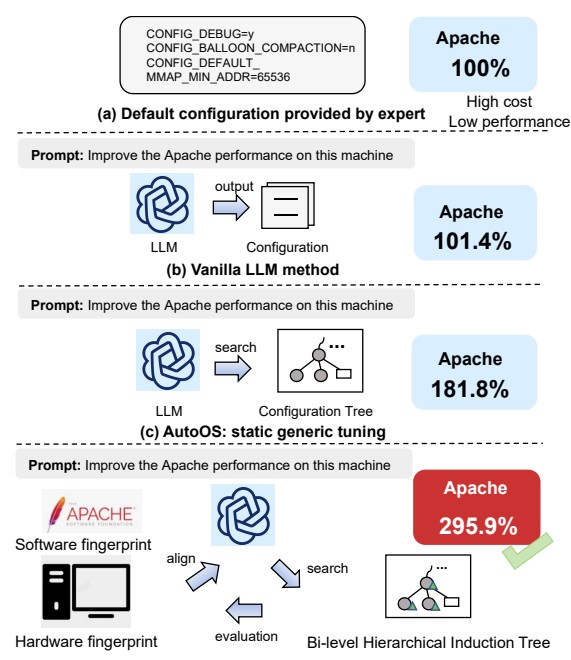

*Figure 1.* QiMeng-PerceptOS achieves superior kernel performance through closed-loop semantic alignment, bridging the gap between low-level hardware/software telemetry and the LLM's high-level reasoning space.

## 1. Introduction

Operating systems (OS) serve as the foundational substrate of modern computing, orchestrating software-hardware interactions through expansive configuration spaces of over 15,000 kernel compile-time parameters (Franz et al., 2021). With the development of computer science, software workloads are increasingly specialized and characterized by long-standing, dedicated instances (e.g., dedicated Redis or Apache instances (Fielding & Kaiser, 2002; Boudreau, 2012; Baudry & Monperrus, 2015; Hatton et al., 2017)), making application-specific kernel optimization a primary frontier in AI for Systems for unlocking the system's maximum performance potential (Gauthier et al., 2001; Jung et al., 2021; Zhang et al., 2024; Zhou et al., 2025; Kuo et al., 2020).

---

[1]State Key Lab of Processors, Institute of Computing Technology, Chinese Academy of Sciences, Beijing, China [2]University of Chinese Academy of Sciences, Beijing, China [3]Hangzhou Institute for Advanced Study, UCAS, Hangzhou, China [4]Intelligent Software Research Center, Institute of Software, CAS, Beijing, China. Correspondence to: Ling Li <liling@iscas.ac.cn>, Yunji Chen <cyj@ict.ac.cn>.

*Proceedings of the 43rd International Conference on Machine Learning*, Seoul, South Korea. PMLR 306, 2026. Copyright 2026 by the author(s).

However, achieving peak efficiency for a specific application remains a formidable challenge due to the combination of ultra-high dimensionality, prohibitive evaluation costs (40–70 minutes per compilation and test), and extreme system fragility, where subtle misconfigurations often render the OS unbootable (Oh et al., 2021). These factors render traditional methods ineffective. Specifically, bayesian optimization fails under high dimensionality, dependency constraints and boot failure, while cost models and reinforcement learning suffer from poor optimization efficiency due to the prohibitive sample sizes and interactions required for such high-cost environments (Chen et al., 2024). More details are provided in App. C.

While Large Language Models (LLMs) have recently made automated kernel tuning feasible by leveraging its pre-trained knowledge as a strong prior while avoiding fatal options, existing methods primarily follow a static, open-loop paradigm (Chen et al., 2024; Lin et al., 2025a). They treat the kernel tuning as a "black box", relying on pre-defined rules and heuristic search while failing to perceive the dynamic, application-specific runtime behaviors required for extreme optimization. We identify the primary bottleneck as an inherent *Semantic Mismatch* between the LLM's generalized reasoning and the low-level behaviors specific to diverse hardware and software environments. Specifically, this mismatch manifests as three systematic obstacles:

**(1) Perception-alignment dilemma due to blindness to real-world telemetry.** Unlike expert tuning which correlates telemetry with high-level logic, LLMs remain "blind" to real-world physical states like hardware architecture and software behaviors, leading to environment-agnostic and suboptimal configurations.

**(2) Kernel configuration fragmentation under context constraints.** The massive search space often exceeds the context windows of LLMs, forcing current approaches to resort to localized chunking that irreversibly loses critical performance options hidden in deep subtrees.

**(3) Prohibitive evaluation costs compounded by hallucination-induced search drift.** OS kernel evaluation is exceptionally resource-intensive, requiring 40–70 minutes per iteration. Unlike software domains with rapid feedback, this high-cost environment leaves no room for error. Consequently, even infrequent stochastic hallucinations (e.g., proposing invalid parameters or spurious correlations) trigger severe search drift, wasting critical evaluation budgets and hindering optimization efficiency.

To address these challenges, we offer a key insight: the path to optimal tuning lies in a paradigm shift from static open-loop search to a dynamic, closed-loop perception framework. A naive method would be to let the LLM compare numerical performance changes and configuration edits across iterations to help the optimization. However, this faces severe attribution difficulties: specially in hyper-dimensional spaces, performance changes result from the collective impact of many options, making it nearly impossible to work. This still remains a shallow perception that fails to capture actual systemic changes. We propose the shift is grounded in establishing a Deep Semantic Alignment between raw low-level telemetry and high-level LLM reasoning. We introduce QiMeng-PerceptOS, an autonomous framework that instantiates this semantic alignment paradigm into three collaborative modules: (1) a **Perception Module** that distills unstructured hardware-software telemetry into high-fidelity semantic fingerprints, aligning low-level telemetry with LLM reasoning; (2) a **Global Search Module** utilizing a Bi-level Hierarchical Induction Tree (BHIT) to navigate the vast configuration space via a "hierarchical abstraction-and-search" mechanism, aligning high-dimensional configurations with LLM reasoning; and (3) a **Posterior Enhancement Module** that synthesizes historical search trajectories to suppress hallucinations and refine gains to recover potential semantic mismatches between low-level telemetry and high-dimensional configuration spaces.

Our contributions are summarized as follows:

- **Systematic Dissection of LLM-based OS Optimization.** We identify the fundamental "Semantic Mismatch" in existing static paradigms and categorize their limitations into perception deficit, context constraints, and hallucination-induced search drift.

- **A Novel Perception-Driven Tuning Architecture.** We propose the first close-loop framework achieving deep semantic alignment between raw system telemetry and high-level reasoning through deep semantic perception, hierarchical induction search and posterior enhancement.

- **Comprehensive Empirical Validation.** Extensive experiments across diverse hardware (PCs, multi-core workstations), OS versions, and representative applications (e.g., Redis, Apache, AES, PostgreSQL, RAG) using SOTA LLMs (like GPT-4o-mini, DeepSeek-V3.2, and Gemini-3-pro-preview) demonstrate its efficacy. Notably, when optimizing Redis on Ubuntu, QiMeng-PerceptOS achieved 296.6% of default throughput within only 15 iterations (1 day).

## 2. Problem Formulation

In this section, we will introduce the problem definition as follows:

**Kernel Configuration Space**: Following the previous work (Chen et al., 2024), formally defined as $\texttt{tree} = \langle \mathcal{E}, \mathcal{N} \rangle$, where:

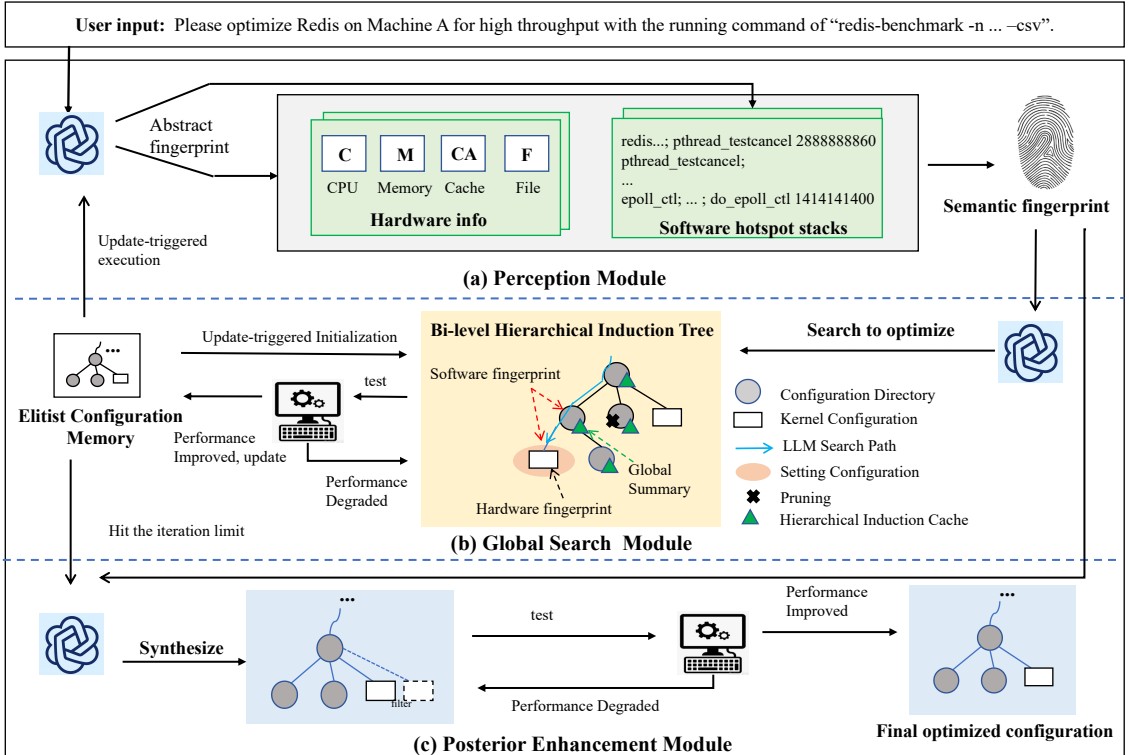

*Figure 2.* **QiMeng-PerceptOS Overview**. The framework orchestrates: (1) **Perception Module**, which performs deep semantic alignment by mapping raw telemetry into software-hardware fingerprints to ground LLM reasoning; (2) **Global Search Module**, which utilizes a Bi-Level Hierarchical Induction Tree (BHIT) to navigate the massive configuration space from a global perspective; and (3) **Posterior Enhancement Module**, which synthesizes breakthrough configurations by verifying trajectories in the Elitist Configuration Memory.

- Non-leaf nodes $n_1 \in \mathcal{N}$ denote *directories*.

- Leaf nodes $n_2 \in \mathcal{N}$ represent *configuration options*, each with multiple configurable states.

- Edges $\mathcal{E}$ denote hierarchical relationships.

The dependency constraints between configurations are managed via a dynamic mechanism provided by Kconfiglib, where subtrees are automatically inserted or deleted based on the selection of the specific nodes to ensure validity.

**Software-Specific Kernel Optimization Task**: Given application $k$, an evaluation metric $f_k \in \mathcal{F}$ (quantified as benchmark score or throughput) maps configurations to performance:

$$\text{score}_k = f_k(\mathbf{tree})$$

The QiMeng-PerceptOS framework seeks an optimal configuration:

$$\mathbf{T}_k^* = \mathcal{A}\left(\mathbf{T}_0, \mathcal{H}, \mathcal{S}_k\right), \quad \text{where } f_k(\mathbf{T}_k^*) = \max_{\mathbf{T}} f_k(\mathbf{T})$$

where $H$ and $S_k$ denote hardware and software information, and $T_0$ is the default configuration. The core complexity stems from the high dimensionality of $k$, requiring a framework that automatically aligns low-level hardware-software features with the LLM's semantic space and efficiently prune the vast configuration space to find $\hat{\mathcal{T}} \approx \mathcal{T}_k^*$ while strictly satisfying the bootability constraint.

## 3. Methodology

### 3.1. Overview

To address software-specific OS kernel optimization, as shown in Fig. 2, QiMeng-PerceptOS comprises three modules: the **Perception Module** (Sec. 3.2) transforms raw telemetry into semantic fingerprints, aligning actual application behavior with LLM reasoning; the **Global Search Module** (Sec. 3.3) employs a Hierarchical Induction Tree guided by a Global Summary Template. It prunes irrelevant subtrees from a global perspective, efficiently locating promising configuration regions and aligning high-dimensional configurations with LLM reasoning; the **Posterior Enhancement Module** (Sec. 3.4) performs posterior synthesis of historical trajectories and recover potential semantic mismatches.

As illustrated in Alg. 1, QiMeng-PerceptOS proceeds in two phases: (i) an iterative optimization loop (L4–29) that

**Algorithm 1** Pseudo code for software-specific kernel optimization algorithm in QiMeng-PerceptOS

1: **Input:** Software $\mathcal{S}$, Initial configuration tree $\theta_0$, Budget $T$
2: **Output:** Optimized configuration $\theta^*$
3: Initial elitist configuration memory $\mathcal{M} = [(\theta_0, 0)]$, Summary memory dict $\mathcal{C}$, Window Hyperparameter $v, k$, Max iterations of global search $n$
4: **for** $t = 1$ **to** $n$ **do**
5:    **if** First iteration or Performance $P_t > best(P\_in\_M)$ **then**
6:       $\mathcal{H} = $ CaptureHardwareInfo()
7:       $\mathcal{H}^* = $ LLM_HardwareTemplate($\mathcal{H}$)
8:       $ST_{hot} = $ GetTopKStacks($\mathcal{S}$) {Sorted by sampling frequency in descending order}
9:       $\mathcal{ST}_{\text{hot}} = ST_{hot}[: v] \cup LLM\_Select(ST_{hot}[v : k])$
10:      $ST^* = $ LLM_SoftwareTemplate($\mathcal{ST}_{\text{hot}}$)
11:    **end if**
12:    $\mathcal{T}_t = $ InitTree($\mathcal{M}[-1]$), $expanded\_set = \{root(\mathcal{T}_t)\}$
13:    **repeat**
14:       Go to the level of $expanded\_set.pop()$
15:       Let $\mathcal{D}$ be child directories and $\mathcal{O}$ be config options at current level
16:       **if** $\mathcal{D} \notin \mathcal{C}$ **then**
17:          $\mathcal{C}[\mathcal{D}] \leftarrow $ LLM_HierarchicalSummary($\mathcal{D}$)
18:       **end if**
19:       Global Vision $VS = \mathcal{C}[\mathcal{D}]$
20:       $expanded\_set.append($LLM_Prune($\mathcal{D}, VS, ST^*$))
21:       $propose\_set = $ LLM_Select($\mathcal{O}, ST^*, \mathcal{H}^*$)
22:       $\theta_t = $ ApplyModification($\mathcal{T}_t, propose\_set$)
23:       $\theta_t = $ MaintainDependency($\theta_t$) {Subtree Inserted or deleted following the previous work}
24:    **until** $expanded\_set = \emptyset$
25:    $P_t = $ Evaluate($\theta_t, \mathcal{S}$)
26:    **if** $P_t > best(P\_in\_M)$ **then**
27:       $\mathcal{M} = \mathcal{M} \cup \{(\theta_t, P_t)\}$
28:    **end if**
29: **end for**
30: **for** $i = 1$ **to** $T - n$ **do**
31:    $\theta_{try} = \theta_0$
32:    **for** each batch of modified options $\mathcal{B} \subset \mathcal{M}$ **do**
33:       $\Delta\theta = $ LLM_TryEnhance($\mathcal{B}$)
34:       $\theta_{try} = $ IncrementalUpdate($\theta_{try}, \Delta\theta$)
35:    **end for**
36:    $\theta^* = $ Eval_and_SelectBest($\theta_{try}, \theta^*$)
37: **end for**

executes the Perception and Global Search modules across several trials, followed by (ii) a posterior enhancement phase (L30–37) that probabilistically composes breakthrough configurations after global search reaches plateaus by auditing the high-performance configurations accumulated in Elitist Configuration Memory $M$.

### 3.2. Perception Module

The fundamental challenge in software-specific kernel tuning lies in the Perception Gap: unlike manual expert prompting, QiMeng-PerceptOS perceives software characteristics directly from raw telemetry to align low-level telemetry with LLM reasoning. However, this data—comprising millions of stack samples and thousands of hardware registers—is high-dimensional and noise-intensive, far exceeding LLM context windows. The Perception Module bridges this gap by distilling unstructured telemetry into a high-fidelity semantic fingerprint $\mathcal{F}$, ensuring subsequent Global Search Module decisions are anchored in precise runtime behaviors rather than abstract heuristics.

**Hardware Template.** QiMeng-PerceptOS autonomously ingests raw hardware descriptors $H$ (e.g., via /proc/cpuinfo, lstopo). To prevent context saturation, QiMeng-PerceptOS employs a hardware template guiding the LLM to use range merging and structural grouping techniques. This distills raw architectural states into a concise summary $H^*$ (see Alg. 1, L6-L7), capturing critical factors such as CPU architecture, cache hierarchies, and memory capacity while preserving the "architectural essence" for optimization with minimal token overhead. The template and an example are provided in App. B.1.

**Diversity-Aware Input Construction for Software Semantic Summarization.** Extracting software fingerprints is the cornerstone of overcoming the semantic generalization gap. Traditional tuning relies on numerical counters (e.g., IPC, cache miss rates); however, LLMs struggle to derive actionable insights from such data due to the lack of direct semantic mapping. Instead, QiMeng-PerceptOS captures the "semantic DNA" of software through hot call stacks and their relative rankings, as shown in Fig. 7. Additionally, QiMeng-PerceptOS employs a diversity-aware windowing strategy to balance representative coverage and semantic variety of stacks. First, the top $v$ stacks are selected based on sampling frequency to form the core set $S_{\text{base}}$ that identifies primary features. Subsequently, for stacks ranked within the interval $[v, v + k]$, an LLM-based semantic discriminator identifies a subset $\Delta S$ containing features semantically distinct from $S_{base}$ yet functionally relevant to optimization (see Alg. 1, L8–L9, and the template in App. B.2). Both $S_{\text{base}}$ and $\Delta S$ are then concatenated as the input of the software template.

**Software Template.** As outlined in Alg. 1 (L10), the selected stack set $\mathcal{ST}_{\text{hot}} = S_{\text{base}} \cup \Delta S$ is processed via a specialized software template as shown in Fig. 2. This guides the LLM to generate dense software fingerprints while minimizing token overhead, adhering to four principles: (1) Structural Abstraction: Truncates framework boundaries to isolate application-kernel interactions (e.g., folding internal runtime boilerplate in Python/ONNX); (2) Symbolic Normalization: Removes uninformative labels (e.g., [unknown]) and utilizes context-aware abbreviations; (3) Horizontal Path Merging: Aggregates similar stacks across threads to eliminate redundancy; (4) Magnitude Scaling: Converts absolute counts into compact ordinal rankings (e.g., $20.9\,\mathrm{G}$, $15\,\mathrm{k}$) to maintain numerical sensitivity. The template and an example are provided in App. B.3.

By adhering to explicit and universal formulation protocols, our templates enable the LLM to consistently extract salient hardware-software features across diverse contexts. This deterministic abstraction endows templates with superior generalizability, ensuring robust performance across heterogeneous environments. We formalize the perception task as a mapping function $\Phi$:

$$\mathcal{F} = (\mathcal{H}^*, ST^*) = \Phi(\mathcal{H}, \mathcal{S})$$

### 3.3. Search Module with Global Vision

To align high-dimensional configurations with LLM reasoning and navigate the exponentially vast configuration space, QiMeng-PerceptOS introduces the Bi-Level Hierarchical Induction Tree (BHIT), a specialized data structure architected upon kconfiglib. Beyond supporting standard atomic operations—such as recursive directory navigation, option configuration, and metadata retrieval—BHIT introduces two core innovations: a Bi-Level Hierarchical Induction and Induction Caching Mechanism.

**Bi-Level Hierarchical Induction**. QiMeng-PerceptOS employs the LLM to perform a Depth-First Traversal across the BHIT, executing pruning and configuration at each directory node (see Alg. 1, L13–L24). Unlike flattened or heuristic pruning, BHIT reconstructs decision contexts by concurrently exposing two layers: immediate configuration options ($O$) and synthesized subdirectory semantic summaries ($VS$). By fusing localized details with subdirectory abstractions, this mechanism enables the LLM to infer downstream functional trajectories without full recursive expansion, significantly reducing the search space.

Semantic summaries are dynamically generated via a **Global Summary Template (GST)**, which guides the LLM to filter low-entropy symbols (e.g., status flags) and reconstruct a subsystem's functional essence based on option descriptions and downstream paths. This process effectively compresses sparse metadata into dense semantic representations. To eliminate computational redundancy across search iterations, we implement an Induction Caching Mechanism at each directory node. Critically, BHIT adopts an on-demand synthesis strategy: summaries are generated only when the LLM first traverses a directory during the search process. This lazy strategy ensures GST-based synthesis occurs only once for any unique subdirectory set, ensuring scalability and drastically improving inference efficiency (see Alg. 1, L16-19). Templates and examples are in App. B.4.

During depth-first traversal, QiMeng-PerceptOS injects runtime fingerprints into reasoning prompts to exploit the LLM's internal knowledge for effective space pruning, while calibrated sampling temperatures maintain exploratory stochasticity. In contrast to prior static template-based approaches, in the Pruning Phase, the global search module correlates the software semantic fingerprint $ST^*$ (e.g., I/O patterns) with hierarchical summaries $VS$ to allow the LLM to prune branches exhibiting semantic misalignment with workload characteristics automatically and efficiently locate promising configuration space without being constrained by heuristic rules. Subsequently, in the Configuration Proposal Generation Phase, the module fuses low-level hardware features $H^*$ (e.g., cache topology) with software fingerprints. This mechanism compels the LLM to seek optimal configurations at the intersection of physical constraints and dynamic software demands (see Alg. 1, L20-21). Templates are provided in App. B.5 & B.6.

**Robust Evolution via Elitist Configuration Memory.** QiMeng-PerceptOS maintains an Elite Configuration Memory ($M$) to persist optimal configuration-performance tuples, establishing a closed-loop optimization mechanism. Upon identifying a superior configuration $\theta_t$ at iteration $t$, the system updates $M$ and re-characterizes the semantic fingerprints to reflect the evolved kernel state. This closed-loop mechanism ensures that performance improvements trigger fingerprint updates, which in turn inform subsequent optimization trajectories. By initializing iterations from the latest elite state, the system adaptively tracks the semantic behavioral drift induced by OS state transitions, guaranteeing cumulative performance gains across successive optimization iterations (see Alg. 1, L26-28).

### 3.4. Posterior Enhancement Module

**Posterior Enhancement via Historical Synthesis.** After multiple optimization rounds within the perception-search closed-loop, the performance enters a plateau phase, where extra global close-loop search often introduces additional bad options. To recover possible semantic misalignments between the semantic fingerprints $F$ and the current configuration at this time due to hallucinations/errors in the previous closed-loop, QiMeng-PerceptOS introduces a Posterior Enhancement Module for refined post-hoc tuning to have a chance to mine marginal gains from historical trajectories with a specific probability. Since directly attributing hallucinations or errors of large language models in high-dimensional spaces is difficult, this module serves as trial-and-error detection. As illustrated in Alg. 1 (L30-36), it conducts a batch sequential re-audit of previously modified options by LLM in elite configurations memory $\mathcal{M}$ based on current semantic fingerprints $\mathcal{F}$ to evaluate the alignment between candidates and runtime characteristics. Configurations exhibiting semantic incongruence with $\mathcal{F}$—indicative of possible hallucinations—are filtered. In contrast, conflict-free entries are retained, as their existence in $\mathcal{M}$ serves as a prior of empirical performance gains. This process yields a distilled, high-performance configuration output. (The template is provided in App. B.7).

*Table 1.* Testbed Configurations.

| No. | Hardware (Processor) | Physical Cores | RAM | Swap Space | OS | Linux Version | Num of Configurations |
|-----|---------------------|----------------|-----|------------|-----|---------------|----------------------|
| 1 | PC (AMD Ryzen 9 5900HX) | 8 | 14.92GB | 0 | Ubuntu 22.04 | 6.8.12 | 17211 |
| 2 | PC (Intel Core i7-13620H) | 10 | 15.25GB | 8GB | Fedora 42 | 6.14.11 | 17942 |
| 3 | Workstation (Intel(R) Xeon(R) w7-3455) | 24 | 251.15GB | 8GB | Ubuntu 24.04.2 | 6.11.0-29 | 17576 |

*Table 2.* Evaluated benchmark workloads with their respective benchmarking tools.

| Benchmark | Representative Workload Characteristics | Metric |
|-----------|----------------------------------------|--------|
| **Redis** | Intensive memory management | redis-benchmark |
| **Apache** | I/O scheduling, process management and network stacks | ApacheBench (ab) |
| **AES** | CPU scheduling and hardware acceleration | cryptsetup benchmark |
| **PostgreSQL** | I/O scheduling and memory caching | pgbench |
| **RAG(Chromadb)** | Compute-memory interleaved pattern | throughput |

To transform breakthrough configurations into an executable kernel state, QiMeng-PerceptOS employs a Sequential Incremental Application strategy. Validated items are applied in the order they appeared in the original high-performance set, with Kconfiglib integrated for rigorous dependency validation; items with unsatisfied prerequisites are automatically skipped to ensure internal consistency.

For boot failures despite passing dependency checks, we implement a two-layer safeguard (Chen et al., 2024): the LLM first attempts to localize the problematic options through heuristic reasoning, failing which the system reverts to deterministic binary localization. This coupling of heuristic logic and deterministic verification guarantees the ultimate bootability of the optimized kernel.

## 4. Experiment

### 4.1. Experimental Setup

**Testbed Configurations.** We evaluate QiMeng-PerceptOS across diverse hardware architectures—spanning high-concurrency workstations to different consumer PCs—paired with distinct OS distributions to capture varied kernel philosophies and release cycles (see Table 1).

**Benchmark Workloads and Performance Metrics.** We select five applications as optimization targets: Redis (in-memory store (Carlson, 2013)), Apache (web server (Laurie & Laurie, 2003)), AES Encryption (compute-intensive (Daemen & Rijmen, 1999)), PostgreSQL (relational database (Worsley & Drake, 2002)), and a Naive RAG workload using ChromaDB indexed with the UltraDomain-Art dataset (Wang & Duan, 2025; Qian et al., 2025). These workloads are strategically chosen to encompass a comprehensive and diverse range of interaction patterns within the operating system kernel, as shown in Table 2. For multi-component benchmarks, we employ the geometric mean of the performance improvement ratios across all sub-metrics to ensure robust normalization and provide a consolidated measure of the overall gains. The example of benchmark

is provided in Table 15 of App. F. To evaluate the generalizability of QiMeng-PerceptOS, we also extend our benchmarks to containerized environments; further details are provided in App. E.4. Furthermore, a dedicated discussion on the broader applicability and potential scope constraints such as distributed clusters and neural training of QiMeng-PerceptOS is provided in App. A.

**Baseline.** To rigorously evaluate QiMeng-PerceptOS, we establish a three-tier hierarchy of automated baselines: Default configurations from OS vendors represent generalized expert-tuned standards; Vanilla LLM (See App. B.8); and AutoOS (Chen et al., 2024), the state-of-the-art automated kernel optimization framework, employs a set of static, predefined prompts to enhance capabilities across various OS components. To further evaluate diverse tuning paradigms, we assess two non-automated baselines on a representative platform: BYOS, a recent semi-automated, LLM-based system with knowledge graph, and a modified version of AutoOS by us, which optimizes each application directly rather than enhancing various OS components while preserving the framework's original search configurations. This controlled modification aims to evaluate the performance of existing frameworks when their optimization target is directly the target applications. Due to their semi-automated nature, these two baselines are treated as a targeted comparative study distinct from our primary automated evaluation; details are provided in Table 13 and App. E.5. Notably, traditional methods—such as Bayesian optimization, neural networks, and reinforcement learning—are excluded due to the high-dimensional search space, high evaluation costs, and the risk of boot failures (see Table 8).

**Implementation Details.** We instantiate QiMeng-PerceptOS using three representative LLMs: GPT-4o-mini (8B lightweight), DeepSeek-V3.2 (advanced open-source), and Gemini-3-pro-preview (SOTA reasoning). For a fair comparison, the optimization budget is strictly capped at 15 iterations (see App. E.1 for ablation). While baselines execute 15 refinement rounds, QiMeng-PerceptOS allocates

*Table 3.* Performance optimization results across different environments. Percentages in the table represent performance relative to the default configuration. **Bold** indicates the best performance. The *Improv.* row shows the performance gain of our **QiMeng-PerceptOS** relative to Vanilla and AutoOS baselines respectively.

| Environment | LLM | Setting | Redis | Apache | AES | PostgreSQL | RAG |
|---|---|---|---|---|---|---|---|
| PC (AMD) Ubuntu 22.04 | GPT-4o-mini | Vanilla | 104.8% | 101.4% | 100.4% | 101.1% | 102.0% |
| | | AutoOS | 207.3% | 181.8% | 140.8% | 142.8% | 113.4% |
| | | QiMeng-PerceptOS | **278.5%** | **295.9%** | **154.0%** | **154.5%** | **124.3%** |
| | | *Improv.* | ↑173.7%/↑71.2% | ↑194.5%/↑114.1% | ↑53.6%/↑13.2% | ↑53.4%/↑11.7% | ↑15.6%/↑10.9% |
| | DeepSeek-V3.2 | Vanilla | 226.3% | 142.7% | 101.2% | 130.2% | 116.7% |
| | | AutoOS | 220.9% | 196.4% | 140.6% | 142.7% | 117.2% |
| | | QiMeng-PerceptOS | **294.0%** | **306.3%** | **143.5%** | **157.1%** | **128.4%** |
| | | *Improv.* | ↑67.7%/↑73.1% | ↑163.6%/↑109.9% | ↑42.3%/↑2.9% | ↑26.9%/↑14.4% | ↑11.7%/↑11.2% |
| | Gemini-3-pro-preview | Vanilla | 109.0% | 225.7% | 141.4% | 139.7% | 122.2% |
| | | AutoOS | 264.0% | 254.5% | 150.8% | 157.3% | 118.4% |
| | | QiMeng-PerceptOS | **296.6%** | **323.5%** | **157.0%** | **164.9%** | **128.7%** |
| | | *Improv.* | ↑187.6%/↑32.6% | ↑97.8%/↑69.0% | ↑15.6%/↑6.2% | ↑25.2%/↑7.6% | ↑6.5%/↑10.3% |
| PC (Intel) Fedora 42 | GPT-4o-mini | Vanilla | 108.9% | 116.3% | 105.5% | 100.6% | 111.3% |
| | | AutoOS | 108.2% | 122.5% | 117.7% | 106.5% | 102.2% |
| | | QiMeng-PerceptOS | **147.0%** | **139.7%** | 116.8% | **112.3%** | **133.8%** |
| | | *Improv.* | ↑38.1%/↑38.8% | ↑23.4%/↑17.2% | ↑11.3%/↑-0.9% | ↑11.7%/↑5.8% | ↑22.5%/↑31.6% |
| | DeepSeek-V3.2 | Vanilla | 132.5% | 117.8% | 106.6% | 100.0% | 136.5% |
| | | AutoOS | 121.6% | 137.2% | 111.9% | 107.2% | 109.7% |
| | | QiMeng-PerceptOS | **143.7%** | **147.9%** | **118.4%** | **116.0%** | 133.7% |
| | | *Improv.* | ↑11.2%/↑22.1% | ↑30.1%/↑10.7% | ↑11.8%/↑6.5% | ↑16.0%/↑8.8% | ↑-2.8%/↑24.0% |
| | Gemini-3-pro-preview | Vanilla | 107.3% | 121.5% | 107.1% | 109.7% | 130.5% |
| | | AutoOS | 123.7% | 149.1% | 121.2% | 113.9% | 116.5% |
| | | QiMeng-PerceptOS | **145.6%** | **165.7%** | 118.8% | **115.5%** | **135.5%** |
| | | *Improv.* | ↑38.3%/↑21.9% | ↑44.2%/↑16.6% | ↑11.7%/↑-2.4% | ↑5.8%/↑1.6% | ↑5.0%/↑19.0% |
| Workstation Ubuntu 24.04 | GPT-4o-mini | Vanilla | 100.1% | 102.3% | 102.4% | 104.2% | 102.3% |
| | | AutoOS | 100.3% | 100.0% | 102.3% | 103.4% | 100.3% |
| | | QiMeng-PerceptOS | **123.8%** | **128.9%** | **108.0%** | **109.2%** | **119.7%** |
| | | *Improv.* | ↑23.7%/↑23.5% | ↑26.6%/↑28.9% | ↑5.6%/↑5.7% | ↑5.0%/↑5.8% | ↑17.4%/↑19.4% |
| | DeepSeek-V3.2 | Vanilla | 108.1% | 114.8% | 103.0% | 102.8% | 115.9% |
| | | AutoOS | 100.0% | 100.0% | 101.5% | 107.9% | 100.0% |
| | | QiMeng-PerceptOS | **124.6%** | **133.4%** | 102.6% | **115.6%** | **120.1%** |
| | | *Improv.* | ↑16.5%/↑24.6% | ↑18.6%/↑33.4% | ↑-0.4%/↑1.1% | ↑12.8%/↑7.7% | ↑4.2%/↑20.1% |
| | Gemini-3-pro-preview | Vanilla | 118.4% | 114.6% | 102.2% | 109.3% | 116.9% |
| | | AutoOS | 119.1% | 109.4% | 110.1% | 101.4% | 107.1% |
| | | QiMeng-PerceptOS | **128.9%** | **139.3%** | **112.6%** | **113.4%** | **117.9%** |
| | | *Improv.* | ↑10.5%/↑9.8% | ↑24.7%/↑29.9% | ↑10.4%/↑2.5% | ↑4.1%/↑12.0% | ↑1.0%/↑10.8% |

13 to autonomous search and 2 to Posterior Enhancement, maintaining an identical total budget. For reproducibility, the Search Stage employs a temperature of 1.0 to balance exploration and exploitation followed the previous works (Chen et al., 2024), and an API seed of 47 (the 15th prime: this establishes a deterministic, non-arbitrary link to our 15-iteration budget). The Perception Module utilizes a seed of 0 and temperature of 0 for deterministic fingerprinting ($v = 20, k = 50$ to accommodate the LLM's context window, and the LLM input of fingerprint is constructed by concatenating the core call stack with the top-5 stacks prioritized by the model), while the Posterior Enhancement uses seeds 1 and 2 with a temperature of 1.0. For statistical reliability, we report the mean and standard deviation across independent runs; our method exhibits high stability and low sensitivity to seed variations (see App. E.3).

## 4.2. Experimental Results

**Main Results.** Table 3 compares performance across five benchmarks and diverse environments. QiMeng-PerceptOS consistently achieves superior optimization, significantly outperforming both vanilla and SOTA methods. Notably, in the Ubuntu PC environment, the Gemini-3-pro-preview-backed QiMeng-PerceptOS elevates Apache throughput to 323.5% of default, while the leading SOTA reaches only 254.5%. Given the principle of diminishing returns—where marginal gains are exponentially harder to secure at higher tiers—this substantial margin underscores the formidable optimization potency and scaling potential of QiMeng-PerceptOS.

**Robust Optimization Capability Across Applications, Hardware, and OS Distributions**. We evaluate the generalization of QiMeng-PerceptOS across three critical dimensions: (1) Cross-Application Robustness: As summarized in Table 3, QiMeng-PerceptOS consistently outperforms

*Table 4.* Average search time and LLM cost per iteration for three runs in the global search module on the workstation. (Time in minutes and Cost of API call in dollars)

| LLM | Metric | Redis | Apache | AES | PostgreSQL | RAG |
|---|---|---|---|---|---|---|
| GPT-4o-mini | Time (min) | 24.2 | 33.2 | 25.7 | 34.1 | 22.1 |
| | Cost ($) | 0.22 | 0.33 | 0.23 | 0.30 | 0.17 |
| DeepSeek-V3.2 | Time (min) | 78.2 | 75.1 | 33.1 | 105.2 | 60.9 |
| | Cost ($) | 0.25 | 0.33 | 0.13 | 0.48 | 0.10 |
| Gemini-3-pro-preview | Time (min) | 159.6 | 175.6 | 153.1 | 171.5 | 132.4 |
| | Cost ($) | 3.91 | 4.38 | 3.14 | 4.65 | 3.74 |

*Table 5.* Ablation in Perception Module and Global Search Module for GPT-4o-mini-based QiMeng-PerceptOS optimizing apache on the Ubuntu workstation, showing the performance impact of disabling key components.

| | Perception Module | | Search Module | | Result |
|---|---|---|---|---|---|
| | HW | SW | Hierarchy Summary | Close Loop | |
| Ours | ✓ | ✓ | ✓ | ✓ | **128.90%** |
| Ablation 1 | × | ✓ | ✓ | ✓ | 114.34% |
| Ablation 2 | ✓ | × | ✓ | ✓ | 116.41% |
| Ablation 3 | ✓ | ✓ | × | ✓ | 121.70% |
| Ablation 4 | ✓ | ✓ | ✓ | × | 116.70% |

*Table 6.* Ablation in Posterior Enhancement Module (PE) for GPT-4o-mini-based QiMeng-PerceptOS on the Ubuntu PC. Percentages in the table represent performance relative to the default configuration.

| Setting | Redis | Apache | AES | PostgreSQL | RAG |
|---|---|---|---|---|---|
| 13 search | 260.95% | 295.89% | 151.93% | 149.88% | 124.25% |
| 15 search | 260.95% | 295.89% | 152.19% | 149.88% | 124.25% |
| 13 search+PE | **278.53%** | 295.89% | **154.00%** | **154.50%** | 124.25% |

the SOTA, AutoOS, yielding optimized performance such as 278.5% and 154.0% for Redis and AES on Ubuntu PCs based on GPT-4o-mini. This performance is underpinned by our dynamic perception mechanism, which utilizes semantic fingerprints to align fine-grained execution features with the LLM's semantic space. (2) Cross-Hardware Adaptability: QiMeng-PerceptOS maintains a competitive edge across heterogeneous environments, from AMD/Intel PCs to high-concurrency 24-core workstations. In workstation settings, it surpasses AutoOS by 23.5% for Redis and 5.7% for AES, validating its ability to adapt to varying hardware capacities and core counts. (3) Cross-OS Versatility: While static paradigms like AutoOS exhibit brittleness when migrating to newer distributions—achieving only 102.2% of default performance on Fedora RAG—QiMeng-PerceptOS reaches 133.8%. This gap highlights QiMeng-PerceptOS's superiority in mapping software traits to a pruned configuration space via its Hierarchical Induction Tree.

**Optimization Capability Across diverse LLMs with different capability.** Table 3 details QiMeng-PerceptOS's performance when integrated with LLMs of varying scales. The results demonstrate that QiMeng-PerceptOS significantly amplifies the optimization potential of lightweight models: even when utilizing the 8B-scale GPT-4o-mini, its performance across multiple environments matches or exceeds that of Vanilla and AutoOS methods backed by the Gemini-3-pro-preview. For instance, on the Ubuntu PC, GPT-4o-mini-based QiMeng-PerceptOS improves AES performance to 154% of default, outperforming Gemini-3-pro-preview-based Vanilla (141.4%) and AutoOS (150.8%). Simultaneously, QiMeng-PerceptOS has a chance to elevate the performance ceiling of Gemini-3-pro-preview, boosting Redis throughput to 296.6%, far exceeding the 109% achieved by the Vanilla method. Notably, AutoOS occasionally exhibits "poor performance" when integrated with Gemini-3-pro-preview—such as underperforming Vanilla in workstation PostgreSQL tasks—corroborating our insight that performance bottlenecks stem from semantic mismatch rather than insufficient model knowledge. The discussion of AutoOS's poor performance on workstations is in App. E.6.

### 4.3. Analysis of Cost

QiMeng-PerceptOS automates weeks of expert labor into an hour-scale process. While the perception and posterior enhancement modules incur negligible overhead, the primary bottleneck remains the 40–70 minute kernel evaluation cycle. To mitigate this, we use an asynchronous pipeline that overlaps the LLM-based search for iteration $t + 1$ with the physical benchmarking of iteration $t$, effectively reducing generative latency. For long-term deployments, this one-time overhead is amortized over the service lifecycle, yielding substantial net gains. Table 4 confirms that larger LLMs like Gemini-3-pro-preview incur much higher costs, yet lightweight models like GPT-4o-mini with an inference cost less than $0.40 per iteration enable QiMeng-PerceptOS to achieve strong performance. Notably, QiMeng-PerceptOS demonstrates higher optimization efficiency; detailed optimization curves (Fig. 16) and theoretical analysis are provided in App. D.

### 4.4. Ablation Study

**QiMeng-PerceptOS's superiority arises from the synergistic integration of its components, rather than any isolated feature.** Tables 5 and 6 summarize these experiments based on GPT-4o-mini. Systematic evaluation on the Apache on the workstation (Table 5) quantifies the necessity of each mechanism of the Perception and Global Search modules: excluding hardware fingerprints (Ablation 1), software telemetry (Ablation 2), hierarchical inductive summaries (Ablation 3), or elite configuration memory and closed-loop feedback (Ablation 4) consistently triggers severe performance degradation. Furthermore, As shown in Table 6, experiments substituting the two posterior enhancement rounds with continued global search highlight the

*Table 7.* Impact of Hyperparameter v, k in Perception Module when optimizing OS for Apache on Ubuntu Workstation via GPT-4o-mini-based QiMeng-PerceptOS.

| Setting | $v = 0, k = 0$ | $v = 0, k = 50$ | $v = 20, k = 50$ | $v = 40, k = 80$ |
|---|---|---|---|---|
| **Performance** | 116.4% | 124.6% | 128.9% | 127.5% |

significance of the posterior enhancement module. QiMeng-PerceptOS consistently outperforms baselines even without it. When extra global searches fail, posterior enhancement achieves a marginal breakthrough with some probability especially when performance is already high, confirming its value. When the search phase already finds an optimal, hallucination/error-free config (e.g., Apache/RAG), performance remains unchanged. We report it has a 35% statistical probability of gain across all tests in Table 3 (see App. E.2).

**Hyperparameter Ablation of the Perception Module.** We conduct an ablation study to justify the function of diversity-aware windowing and LLM-based semantic discriminator and study the selection of v, k in the Perception Window. As Table 7 shows: (a) Discriminator as a Safety Net: Moving from $k = 0$ to $k = 50$ (116.4% to 124.6%) with $v = 0$ proves the LLM-based discriminator prevents losing essential call stacks when v is small. (b) Stability via Fixed Windows: $v = 20, k = 50$ outperforming $v = 0, k = 50$ shows that fixed top-$v$ stacks ensure core semantic features are not omitted despite LLM instability. (c) Performance parity between $v = 40, k = 80$ and $v = 20, k = 50$ indicates performance is insensitive to moderate parameter increases for semantic fingerprints effectively shield the LLM from redundant information.

## 5. Related Work

**OS Kernel Optimization**. The OS kernel encompasses runtime options, tunable via various methods (Bergstra et al., 2011; Herzog et al., 2021; Zhou et al., 2024; Hwaish et al., 2025; Fu et al., 2023), and compile-time options. We focus on the latter, a vast space where lengthy recompilation renders traditional approaches impractical (Franz et al., 2021; Xia et al., 2023). Recent works leverage LLMs: AutoOS (Chen et al., 2024) performs open-loop optimization via static prompts but lacks dynamic perception of runtime interactions. Similarly, BYOS (Lin et al., 2025a) assists LLM with expert-curated knowledge graphs; however, this reliance on costly human priors compromises autonomy and fails under sparse documentation. Notably, OS-R1 (Lin et al., 2025b) distills teacher knowledge into open-source lightweight models using GRPO. Yet, this training-heavy paradigm is tied to fixed hardware, kernel versions, and incur performance ceilings bounded by the teacher model. Consequently, OS-R1 represents a distillation technique orthogonal to our perception-driven framework rather than a direct baseline. Overall, while traditional optimization

paradigms (e.g., Neural Networks or Reinforcement Learning) and current LLM-based methods lack environmental perception or confined to numerical sensing, our approach features deep semantic-aware perception.

**Tool-Augmented Methods**. Concurrently, tool-augmented methods have emerged as a powerful paradigm, equipping LLMs (Team et al., 2023; Bai et al., 2023; Achiam et al., 2023; Liu et al., 2024; Zhang et al., 2025) with external tools to perform complex tasks like web browsing and code generation (Mei et al., 2024; Wu et al., 2024; Qiao et al., 2024; Yang et al., 2024; Sridhar et al., 2024; Yue et al., 2025). This approach overcomes the limitations of pure language-based reasoning. However, addressing the challenges of perception and scalability in the specific context of OS kernel optimization remains a critical area that requires in-depth exploration.

## 6. Conclusion

This paper presents QiMeng-PerceptOS, a semantic-aware framework for software-specific OS kernel tuning. Through three synergistic modules, QiMeng-PerceptOS bridges the semantic mismatch inherent in existing methods. Our experiments across diverse workloads highlight QiMeng-PerceptOS's effectiveness and the transformative potential of perception-driven autonomous optimization.

## 7. Limitations and Future Work

QiMeng-PerceptOS achieves significant gains for OS-intensive workloads, yet it has two key limitations: its impact diminishes for compute-intensive tasks with sparse OS interactions, and software-specific optimizations often incur trade-offs for other workloads. Future research will follow four trajectories: (1) Distributed clusters: leveraging federated learning to tailor optimizations for heterogeneous node-specific workloads, thereby maximizing collective cluster performance; (2) Real-time systems: establishing offline pre-optimized libraries for dynamic switching alongside secure fallback mechanisms; (3) Configurable parameter expansion: scaling the framework to a broader range of complex tuning parameters across diverse workloads; and (4) Automated kernel synthesis: exploring autonomous OS code generation, synthesizing high-performance kernel primitives that precisely satisfy hardware-software co-design requirements. See detailed discussion in App. A.

# Acknowledgements

We would like to express our sincere gratitude to everyone who contributed to this work during the submission and rebuttal processes. We are also grateful to the reviewers for their constructive feedback, which enhanced the quality of the manuscript.

This work is partially supported by the NSF of China (Grants No.62525203, 62302483, U22A2028, 92364202), Strategic Priority Research Program of the Chinese Academy of Sciences (Grants No.XDB0660300, XDB0660301, XDB0660302), CAS Project for Young Scientists in Basic Research (YSBR-029) and Youth Innovation Promotion Association CAS.

# Impact Statement

This paper presents work whose goal is to advance the field of Machine Learning. There are many potential societal consequences of our work, none which we feel must be specifically highlighted here.

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

# A. Detailed Discussion

QiMeng-PerceptOS excels in OS-intensive scenarios but faces limitations in two specific contexts:

(1) Low OS-dependency in compute-intensive workloads: The kernel ceases to be a performance bottleneck in this case. Such workloads typically exhibit sparse system calls and shallow call stacks, leaving minimal headroom for OS-level scheduling. To validate this, we applied QiMeng-PerceptOS to neural network training tasks, specifically VGG-16 fine-tuning and LoRA training of small-scale LLM. Experimental results show that the end-to-end training time remains almost unchanged when using Vanilla LLM, AutoOS, or QiMeng-PerceptOS. The root cause of this convergence lies in the bottleneck distribution: although the training process involves periodic OS interactions, operations such as data loading and synchronization account for a negligible fraction of the total execution time. The core bottleneck remains the sustained arithmetic throughput of GPU/CPU compute units. Because QiMeng-PerceptOS targets OS-level resource allocation rather than raw computational kernels, its impact in purely compute-intensive scenarios has inherent limitations. Nevertheless, the increase in OS-intensive workloads—such as network services, databases, and memory management frameworks—presents an expanding optimization space for QiMeng-PerceptOS. To delineate the boundary of applicability, Table 2 provides a structured reference of classic interaction patterns with the OS in OS-intensive workloads. Furthermore, while focused on the kernel, QiMeng-PerceptOS's insight is transferable to other high-dimensional, high-cost, high-risk scenarios, such as optimizing other complex systems' compile-time configuration to accelerate workloads that depend on it, which is left to future work.

(2) Inherent multi-dimensional trade-offs: Optimizing OS for a specific application may lead to performance degradation in others, as different workloads have distinct kernel requirements. How to define aggregate metrics for multi-app joint optimization to balance cross-workload effects is a matter for future research. However, QiMeng-PerceptOS prioritizes optimizing the OS for a single application in dedicated environments—a scenario of significant practical value, as core services are typically deployed on dedicated nodes for years.

Although current evaluations focus on single-node performance, QiMeng-PerceptOS demonstrates significant scaling potential in distributed environments. Modern clusters primarily adopt a master-worker architecture, where the master node dispatches requests to specific worker nodes. Since users ultimately communicate with these independent worker nodes, the single-node efficiency enhanced by QiMeng-PerceptOS directly translates into increased aggregate cluster throughput. By augmenting the processing capacity of each worker, QiMeng-PerceptOS establishes a foundation for horizontal scaling. Given the high experimental costs and hardware coordination complexity of large-scale deployments, a full distributed evaluation is a core objective for future work.

# B. Details of Template

## B.1. Hardware Template

---
**Hardware Template**

{Raw Hardware Message} Now you need to compress the arch info concisely to reduce the number of tokens. Please apply: 1) Hierarchy summarization 2) Value range merging 3) Category grouping. Express in a few concise lines. Ensure clarity. Output directly in brief English without other words:

---

*Figure 3.* Hardware Template

As shown in Fig. 4 , QiMeng-PerceptOS automatically acquires and processes raw hardware descriptor information (e.g., by reading data from /proc/cpuinfo and lstopo). To effectively prevent large language models from context overload, QiMeng-PerceptOS utilizes a hardware template that employs two core techniques—range merging and structural grouping—to efficiently compress complex hardware architectural states into a highly concise summary $\mathcal{H}^*$.

## B.2. Diversity-Enhanced Sampling Template for Hot Stacks

---
**Diversity-Enhanced Sampling Template for Hot Stacks**

(Round 1)
{The content of the stacks in the window[v:k]} This is the top v-k call stacks (by time proportion). Now, there are many identical call stacks. Remove the duplicates, and then output the deduplicated call stacks in the format I provided. Do not output any other text

---

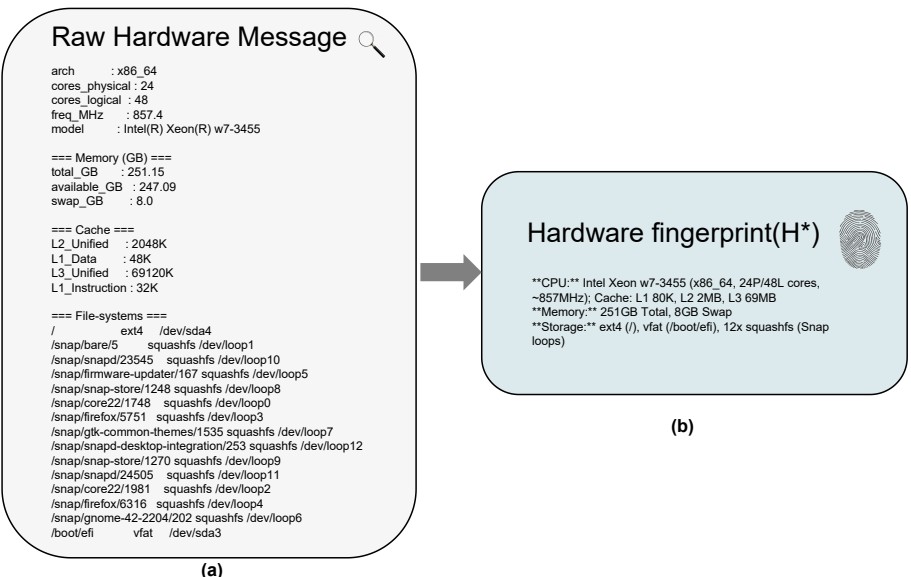

*Figure 4.* Examples of (a) the hardware information obtained originally on workstation in the experiment and (b) the Hardware fingerprint processed by the LLM through the hardware semantic template.

(Round 2)
This is a partial call stack (by time proportion). I now aim to optimize the program's throughput by optimizing the kernel. Please analyze the relationship between each call stack and the OS kernel in sequence. Analyze them according to numbers 1, 2... 30, then summarize the types of relationships between the call stacks and the OS, and provide the most classic call stack for each type, formatted as 'Type 1: stack; Type 2: stack':

(Round 3)
Based on the previous analysis, now only output 'Type: complete representative stack time number', do not output any other text

(Round 4)
{The content of the fixed-priority core set $S_{\text{base}}$}} This is the program's top 20 call stacks (by time proportion)." + "These are the representative types and call stacks from the program's call stacks ranked 20-50, which may contain duplicates of the types found in the top 20" + {LLM output of round 3} + "Remove any types that are identical to those in the top 20 call stacks. Then, analyze each remaining type to determine whether it is identical or different from the types represented in the top 20. Finally, output the representative call stacks from the filtered 20-50 list, using the same call stack format provided. Do not output the types. If a type is entirely identical to one in the top 20, analyze and output which one it matches, then output None. Types with slight variations may be retained. Now, please analyze and then output:

*Figure 5.* Semantic Template for Constructing Heterogeneous Call-Stack Set $\Delta S$

## B.3. Software Template

### Software Template

(Round 1)
{Raw Telemetry Call Stacks} This is my program's call stack. I now want to use an LLM to modify menuconfig configurations to optimize the program's throughput. Therefore, I need to first remove redundant information from this call stack to reduce the number of tokens and minimize context. Please organize the call stack information according to the following guidelines:

1. Library function compression: [xx];[xx]→[xx]×2

2. Framework boundary truncation: python3;[...10 Python internal functions, details irrelevant for OS optimization...];[onnxruntime.so] → py3;[critical detail];[onnxrt.so]

3. Similar call stack merging: Thread 1: python3;abc;␣␣xxx 100 & Thread 2: python3;abc;␣␣yyy 200 → python3;abc;␣␣xxx (100) —␣␣yyy (200)

4. Abbreviation for frequently occurring common words: When every call stack starts with python3, specify py3: python3 at the very beginning, then change the start of each subsequent call stack to py3. (Note: This is an example. Apply this to any other frequently occurring words that can be abbreviated without causing confusion.)

5. For numbers with suffixes like 'k' at the end of call stacks, use abbreviations to reduce token usage.

6. Retain keywords crucial for identifying bottlenecks and system status, ensuring precise localization even after significant trimming.

7. remove [unknown], simplify overly long function names.

8. "Retain" the critical sys call path and essential details for each operation

Please now output the call stack adhering to the above criteria:

(Round 2)
The previous simplification may have been too aggressive. Rethinking and retain the critical words and essential details in stack. Now output again without other words:

*Figure 6.* Software Template

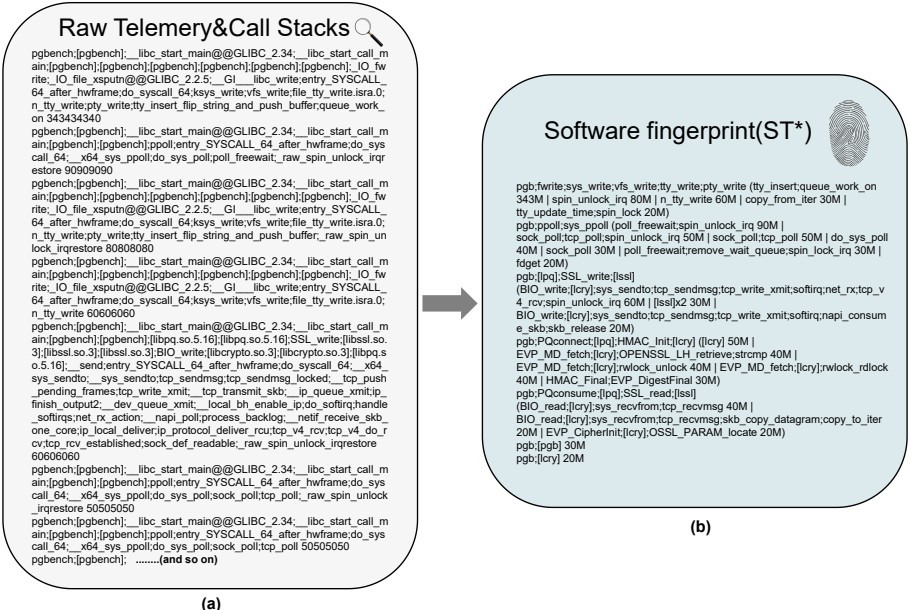

**(a)**  **(b)**

*Figure 7.* Example of (a) the top several dozen call stacks of PostgreSQL's pgbench on workstation, which are sorted by execution time within a sampling window, with the numeric suffix in each row indicating the hit count of the software counter and (b) the runtime fingerprint of PostgreSQL in this environment after processing through the software template.

As illustrated in Fig. 7, QiMeng-PerceptOS applies an optimized software template to the final selected hot call stack set $\mathcal{ST}_{\text{hot}} = \mathcal{S}_{base} \cup \Delta\mathcal{S}$, aiming to eliminate redundancy and prevent useful information from being obscured. The template adheres to the following four compression principles: (1) Trimming framework boundaries, (2) Removing invalid labels, (3) Aggregating similar stack paths across multiple threads, and (4) Scaling numerical magnitudes.

### B.4. Global Summary Template

**Global Summary Template**

{The content of directory node which is the child of the current node} This is the information from the subdir, which I need to input as context for the LLM. Currently, it's taking up too many tokens, so I want to reduce the information. Now, please keep only the directory descriptions, remove the serial numbers, statuses, and the names inside (). Then, based on the directory descriptions, create a concise summary without including the status, using a few lines to briefly describe it, with fewer tokens. Now, please output summary directly.

Ensure clarity. Output directly in brief English without other words:

*Figure 8.* Global Summary Template

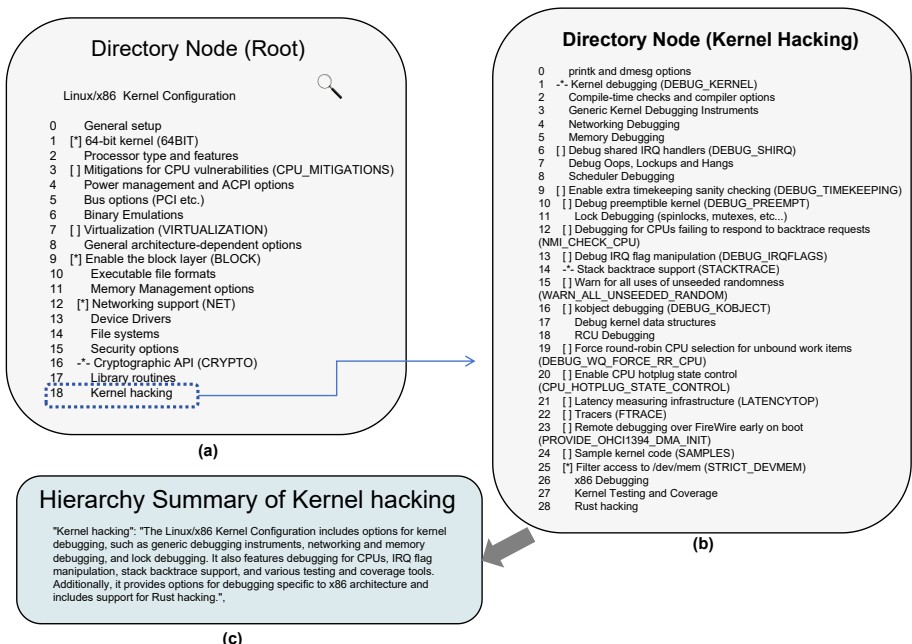

*Figure 9.* Examples of Hierarchical Summary Generation

Fig. 9(a) illustrates an example of hierarchical summary generation when the Large Language Model (LLM) is positioned at the root node during the initial stage of a depth-first traversal of the configuration tree. At this phase, the hierarchical summarization mechanism sequentially generates descriptive summaries for all subdirectories under the root node. Taking the "Kernel Hacking" subdirectory in Fig. 9(b) as an example, the LLM utilizes a global summary template to refine and extract the contents of that directory. This process produces the global context-aware summary shown in Fig. 9(c), which is subsequently stored in the cache dictionary.

### B.5. Pruning Template when Depth-first Traversal

**Pruning Template**

(Round 1: Analyze)
**{Hierarchy Summary for all the sub-directory nodes}** I want to explore the configuration of the Linux kernel's 'config' to target. I will give you the top k hot call stacks obtained from the perf program." + **{software fingerprint}** + f"I will sequentially show you each level of menuconfig's directories, and I need you to tell me which directories(Do not tell me sub directories, only the father directories) are possibly related to target_object based on your existing knowledge.
# Here's how I'll show you the directories which are in some level in menuconfig:[number] [directory name] , please analyze. {the content of the sub-directories in the current node}'

(Round 2: Output in format)
I want to explore the configuration of the Linux kernel's 'config' to {target}. I will sequentially show you each level of menuconfig's directories, and I need you to tell me which directories are possibly related to {target_object} based on the previous conversation. Here's how I'll show you the directories which is in some level in menuconfig:[number] [directory name]  Your response format: For relevant directories: [number] [directory name]  For example, when I give:0 memory setting(mem) 1 computer name(Name) your answer :' 0 memory setting(mem) ' because the memory setting is related to target but the name is not.  The name in the response should be complete, which is the same as I gave you; for example, if the menu name is 'IP: TCP syncookie support (SYN_COOKIES)', it should not only show 'TCP syncookie support (SYN_COOKIES)', it must show 'IP: TCP syncookie support (SYN_COOKIES)'. No extra explanations needed. Do not recommend any directory related to Soc selection,device driver, platform type directories. Do not mention reason. When the number of directory is one, do not output sub directory.Please obey the rules. Here are some directories,please

recommend:

*Figure 10.* Pruning Template

## B.6. Configuration Select Template when Depth-first Traversal

**Template for Binary Configuration Options**

(Round 1: Initial Analysis)
{Options in this level} For {target} , analyze each of the following settings separately to determine whether they will increase or decrease {target_object} if the setting(the following settings) is enabled:

(Round 2: Aligning software-hardware-aware fingerprints with configuration space options)
I will give you the top k hot call stacks obtained from the perf program. Based on the stacks, analysis and experience above, provide the options that could potentially affect {target_object} and analyse whether enabling the settings will increase or decrease target_object while ensuring successfully booting up OS after modification.Analyze while considering the hardware I use."+**{hardware fingerprint}**+".The stacks are as follows:"+ **{software fingerprint}** +"The options are: {Options in this level}"

(Round 3: Output in format)
According to the above analysis, for the options that could potentially impact {target_object} , determine whether each option will increase or decrease {target_object}, Output format: 'increase: Option Name1 Option Name2 decrease: Option Name1 Option Name2'. No explanation, no extra useless words.
For examplewhen related option is IO Schedulers (IOS) DYNAMIC_DEBUG(DD) , the analysis is that IO Schedulers (IOS) will increase the score and DYNAMIC_DEBUG(DD) will decrease the score your answer : 'increase: IO Schedulers (IOS) decrease: DYNAMIC_DEBUG(DD) .Output complete name ,for example,output 'IO Schedulers (IOS)',do not output 'IOS' only! Complete name is important, you need attention. In the output, the assessment of enabling each option should align with the previous analysis, indicating whether it will increase or decrease {target_object}. Do not mention reason.Do not output options about Peripheral drives, just ignore.And skip the options which may lead to boot failure. The option names should maintain consistent capitalization!!!. Please obey the rules. Output complete name as I said. please output:

*Figure 11.* Template for Binary Configuration Options

**Template for Numerical Configuration Options**

I'm exploring the Linux kernel's menuconfig for configurations that might {target}. I will give you the top k hot call stacks obtained from the program."+ **{software fingerprint}** +"Analyze while considering the hardware I use." + **{hardware fingerprint}** + f"Here are multiple numeric options in menuconfig.I have given you the range of each option value in the information above. Please set the options at a time to potentially {target}.
If the option is not related to {target_object}, then remain the default value. My format:[option name] (default value) Your response format: [option name] (recommended value) for example: when I give: "maximum CPU number$(1 \to 2, \ 2 \to 4)$ (cpunum) (1)",your answer is 'maximum CPU number$(1 \to 2, \ 2 \to 4)$ (cpunum) (2)' . Because when the CPU number is more, the speed is usually better. Remember to recommend settings to possibly {target_object} for each option: No extra explanations needed. Only suggest options which maybe {target_object}. Do not mention reason. Do not add units near number in the output.Please obey the rules. Here are some numeric options ,please recommend: {Options in this level}

*Figure 12.* Template for Numerical Configuration Options

**Template for Single-Choice Configuration Options**

I'm exploring the Linux kernel's menuconfig for configurations to {target}. I will give you the top k hot call stacks obtained from the perf program."+ **{software fingerprint}** +"Analyze while considering the hardware I use." +**{hardware fingerprint}** +f"Here are multiple 'select one option' choices in menuconfig. Please select one suitable option at a time to potentially {target_object}. My format:[option1 name]  [option2 name] .. Your response format: '[recommended option name]'
for example: when I give: 'receive bufferrbuf, log buffer(lbuf) /// CPU schedule(cs) ,CPU default(cd) /// SLAB (SLAB) , SLUB (Unqueued Allocator) (SLUB)'. This means there are three 'select one option' choices, and considering {target} ,your answer is :receive buffer(rbuf)CPU schedule(cs) SLUB (Unqueued Allocator) (SLUB) Remember to choose the recommended setting to possibly {target_object} for each option: No extra explanations needed. Only suggest options which may be related.  Do not mention reason.Output complete name ,for example,output CPU schedule(cs),do not output cs only! Complete name is important,you need attention.Please obey the rules. Here are some 'select one option' choices,please choose: {Options in this level}

*Figure 13.* Template for Single-Choice Options

### B.7. Template in Posterior Enhancement Module

> **Template in Posterior Enhancement Module**
>
> My hardware is **{hardware fingerprint}**
> **{software fingerprint}** This is the top k call stacks of the program I'm optimizing. I want to enhance {target_object}. I tried adding the following options to the .config. {config_list}
> I am optimizing the Linux kernel configuration to improve {target_object} and have added the aforementioned options to the .config file. Please analyze one by one which potentially unfavorable options which I added may reduce throughput and cause bad effects. When analyzing, please analyze the direct impact of the configuration. Assuming security is not a concern. Now analyze considering my hardware and stack, and then output the configuration I need to remove to enhance throughput.
> **\*\*Output Requirements\*\*:**
> After your analysis, you must strictly follow the format below to output the final removal list. Please start your final list with the marker ' JSON Result:', followed by a standard JSON list of strings (the configuration items to remove).
> Example format: ... (your analysis) ...
> ### JSON Result: ["CONFIG_AAA=y", "CONFIG_BBB=n"]
> If no items should be removed, output an empty list:
> ### JSON Result:
> []

*Figure 14.* Template in Posterior Enhancement Module

### B.8. Template for the Vanilla LLM Method

> **Template for the vanilla LLM method**
>
> I want to increase the throughput of the software name. How can I modify the configuration options of the linux kernel in menuconfig? Please directly provide specific options and give the following recommended Settings:
> Note: I will add these configuration items at the end of the configuration file, and then use the 'make menuconfig' command to override the previous configuration items. Just enable =y or =n instead of writing CONFIG_XXX is not set. Other types of configuration items can be generated, such as those of numerical types. You can summarize the recommended projects for me at the end.
> ##IMPORTANT OUTPUT FORMAT REQUIREMENTS: 1. Provide your analysis first. 2. Do not use markdown code blocks (no "') for the configuration list. 3. Do not use bullet points inside the configuration list. 4. Strictly output one configuration per line inside the list. 5. You MUST wrap the final configuration list between the tags`<<<CONFIG_START>>>` and `<<<CONFIG_END>>>`. 6. Do not include any explanations inside the tags. 7. Do not repeat any configuration option. Ensure all items are unique.

*Figure 15.* Template for the vanilla LLM method in the experiment

## C. Reasons for the Inapplicability of Traditional Methods

*Table 8.* Reasons for the Inapplicability of Traditional Methods in Software-Specific Kernel Optimization Tasks

| Methodology | Primary Limitations |
|---|---|
| Random Search | Fails to guarantee system bootability; extremely low efficiency. |
| Bayesian Opt. (BO) | Dimension disaster ($D < 300$ vs. $15,000+$); Fails to guarantee bootability and dependency constraints. |
| Reinforcement Learning | Prohibitive sampling costs (40-70 min/iter); Retraining is required for each new software. |
| Cost-Model CNNs | Lack of large-scale labeled datasets; Retraining is required for each new software. |
| Monte Carlo | Prohibitive sampling costs (40-70 min/iter). |

## D. Theoretical Analysis

### D.1. Time Complexity

QiMeng-PerceptOS employs a depth-first traversal strategy over the kernel configuration tree to iteratively adjust settings. Theoretically, this approach maintains the baseline time complexity of $O(n)$ per build-test cycle, where $n$ denotes the total number of configuration items. However, practical performance is influenced by the integration of Perception Fingerprints and Hierarchical Summaries. These components expand the input context, significantly increasing the token count per Large Language Model (LLM) inference. This introduces a constant computational overhead $c > 1$, effectively scaling the wall-clock time complexity to approximately $O(cn)$.

To mitigate this inference latency, QiMeng-PerceptOS incorporates an intra-directory parallelization mechanism. Within a given directory node, the processes of directory pruning and option recommendation are decoupled and executed concurrently. This parallelism effectively amortizes the temporal overhead introduced by the augmented semantic context (fingerprints and

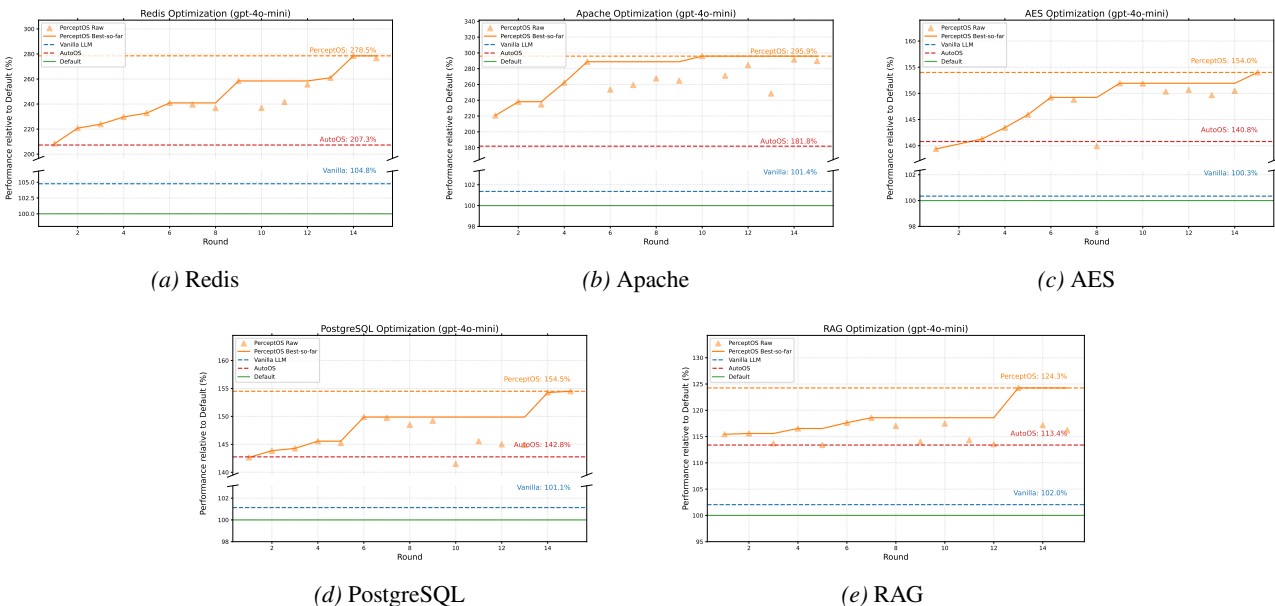

*Figure 16.* **Performance evolution across five diverse applications on the Ubuntu PC optimized by GPT-4o-mini-based QiMeng-PerceptOS.** The horizontal axis represents the search rounds, while the vertical axis indicates the performance of applications after optimization compared to the default.

summaries), ensuring that the system remains efficient without altering the fundamental linearity of the search algorithm.

### D.2. Convergence and Search Efficiency

Unlike traditional optimization methods that target asymptotic convergence over infinite horizons, QiMeng-PerceptOS addresses a budget-constrained optimization problem. It prioritizes maximizing search efficiency to achieve significant performance gains within a limited iteration budget.

As shown in Fig. 16, QiMeng-PerceptOS exhibits significantly higher optimization efficiency than vanilla LLM reasoning and AutoOS, achieving superior performance in substantially fewer rounds.

The search mechanism is grounded in the Deep Semantic Alignment between the software's runtime behavior and the configuration semantics. QiMeng-PerceptOS leverages both the LLM's intrinsic knowledge and the semantic fingerprints to perform pruning of the global search space automatically. Crucially, the system employs a dual-strategy to balance the Exploration-Exploitation trade-off: (1) Exploitation: The LLM's semantic reasoning capabilities guide the search towards high-probability configuration subspaces based on prior knowledge and current semantic context. (2) Exploration: The stochasticity inherent in the decoding process, modulated by temperature scaling, introduces controlled diversity. This prevents the agent from prematurely converging to local optima by encouraging the exploration of previously unvisited configuration regions.

Empirical observations confirm that QiMeng-PerceptOS exhibits high reproducibility and stability, consistently reaching high-performance configuration regions within the limited budget, as seen in Table 11.

## E. In-depth Empirical Analysis and Comparative Studies

### E.1. Ablation on Iteration Budget

To justify the selection of an iteration budget of 15, we conducted a sensitivity analysis to balance computational overhead against incremental performance gains. We extended the GPT-4o-mini-based optimization process from 15 to 24 iterations for Apache on the Fedora PC platform. As demonstrated in Table 9, all evaluated methods—including the Vanilla LLM, AutoOS, and QiMeng-PerceptOS—exhibited marginal improvements beyond the 15th iteration, suggesting that the systems had largely reached their performance plateaus. Consequently, a budget of 15 iterations represents an optimal Pareto frontier between exploration efficiency and tuning efficacy, providing sufficient headroom for convergence across diverse workloads.

### E.2. Further Analysis of the Posterior Enhancement Module

**Mechanism and Case Studies.** The posterior enhancement module is designed to recover potential semantic misalignments when global search reaches a plateau. The module tries to recover from 2 typical hallucination types: obvious contradictions

*Table 9.* Impact of increased iteration budget on optimization performance of Apache on the Fedora PC. The consistency of results between 15 and 24 iterations demonstrates that all methods, including QiMeng-PerceptOS, reach a performance plateau within the designated budget.

| Iterations | 15 | 24 |
|---|---|---|
| Vanilla | 116.3% | 116.3% |
| AutoOS | 122.5% | 122.5% |
| QiMeng-PerceptOS | 139.7% | 139.7% |

*Table 10.* Posterior Enhancement Module Performance Gains Across Different Applications, Hardware and LLMs in table 3.

| Application | Hardware | LLM | Gain |
|---|---|---|---|
| Redis | Ubuntu PC | 4o-mini | **+17.6%** |
| | Fedora PC | DeepSeek | **+1.5%** |
| | Fedora PC | Gemini | +0.6% |
| | Workstation | 4o-mini | **+3.8%** |
| | Workstation | DeepSeek | **+3.7%** |
| AES | Ubuntu PC | 4o-mini | **+2.1%** |
| | Ubuntu PC | DeepSeek | **+1.4%** |
| | Workstation | Gemini | +0.7% |
| PostgreSQL | Ubuntu PC | 4o-mini | **+4.6%** |
| | Fedora PC | 4o-mini | **+6.0%** |
| | Fedora PC | DeepSeek | **+8.3%** |
| | Fedora PC | Gemini | **+6.3%** |
| | Workstation | 4o-mini | **+3.5%** |
| RAG | Fedora PC | 4o-mini | +0.2% |
| | Workstation | DeepSeek | **+2.7%** |
| | Workstation | Gemini | **+1.5%** |

and high-entropy oscillations. For example, in global search stage, the final AES configuration on ubuntu PC modified 331 options compared to the default by GPT-4o-mini-based QiMeng-PerceptOS. The second audit of Posterior Enhancement Module attempted to filter 34 of these options, leading to further performance gains, including: (1) CONFIG_BUG was enabled in iteration 4, via obvious flawed reasoning ("debug helps identify issues"), which contradicts the semantic fingerprint and the target. In the posterior enhancement phase, the LLM correctly identified its overhead and filtered it. (2) the history of CONFIG_UCLAMP_TASK in the elite set was $n \rightarrow y \rightarrow n$, indicating high-entropy oscillations. The LLM re-proposed that it would degrade performance, restoring it to $y$ for test.

**Statistical Results.** In Sec. 4.4, we report a 35% statistical probability of gain across all tests, from 45 scenarios in Table 3 (3 LLMs × 5 workloads × 3 hardware). Here we provide the detailed breakdown. Table 10 shows the 16 cases (out of 45) where Posterior Enhancement Module produced measurable gains ($> 0\%$), yielding a success probability of 16/45 = 35.6% in this set, while others likely reached optima in global search.

While the Apache workload did not exhibit gains in this specific set, subsequent evaluations demonstrate the module's robustness. For instance, in the workstation Docker environment (GPT-4o-mini) in Table 12, performance of Apache increased from 118.0% to 120.6% (+2.6%) with the addition of posterior enhancement. To investigate this further, we conducted additional experiments on the AMD-based Ubuntu PC using O1-powered QiMeng-PerceptOS. In this scenario, the posterior enhancement again yielded consistent improvements: performance improved from 309.5% to 315.2% (+5.7%). Notably, the performance gains presented in Table 10 were achieved when the performance had already reached a high plateau (i.e., at the bottleneck of global search). Following the law of diminishing returns, a further improvement of 2%–10% at this advanced stage constitutes a substantial advancement.

### E.3. Statistical Reliability and Reproducibility Analysis

To evaluate the reproducibility of QiMeng-PerceptOS, we conducted two additional independent trials for each of the five representative applications on the Ubuntu PC platform using GPT-4o-mini-based QiMeng-PerceptOS. To introduce controlled stochasticity, we adjusted the random seeds by adding offsets of 10 and 20 to the baseline seed, respectively. As shown in Table 11, the final optimization outcomes across these three independent 15-iteration sessions exhibit negligible variance, confirming that QiMeng-PerceptOS's performance gains are not artifacts of seed selection.

*Table 11.* Experiment results which show the robustness and reproducibility of QiMeng-PerceptOS . Performance gains (relative to default configuration) across three independent trials with varied random seeds. The negligible standard deviation ($StdDev$) underscores the framework's stability in high-dimensional optimization.

| Trial | 1 | 2 | 3 | Mean | Std Dev |
|---|---|---|---|---|---|
| Redis | 278.5% | 288.8% | 279.5% | 282.3% | 5.73% |
| Apache | 295.9% | 299.3% | 294.2% | 296.5% | 2.63% |
| AES | 154.0% | 150.7% | 152.4% | 152.4% | 1.65% |
| PostgreSQL | 154.5% | 155.1% | 155.0% | 154.9% | 0.32% |
| RAG | 124.3% | 123.5% | 124.1% | 124.0% | 0.42% |

*Table 12.* Relative throughput of Apache running within a Docker container on workstation after optimization, compared to the default configuration. The results demonstrate QiMeng-PerceptOS's capability to navigate additional abstraction layers and generalization in Containerized Environments.

| Method | Relative Performance |
|---|---|
| Vanilla | 101.57% |
| AutoOS | 101.11% |
| QiMeng-PerceptOS | 120.63% |

This high degree of stability is a direct result of our closed-loop, semantic-aware framework. We attribute this stability to QiMeng-PerceptOS's ability to effectively prune the vast parameter space by real-time software-hardware fingerprints and a global system perspective.

### E.4. Performance Generalization to Containerized Environments

To further validate the robustness of the framework across diverse deployment paradigms, we evaluated QiMeng-PerceptOS on an Apache server running within a Docker container on the workstation. Containerization inherently introduces resource isolation overhead and additional abstraction layers for networking and storage, which typically lead to a throughput degradation compared to bare-metal execution. However, as illustrated in Table 12, QiMeng-PerceptOS successfully identified critical kernel parameters that mitigate these isolation bottlenecks, achieving a 20.63% improvement in throughput. This result confirms that even when application behaviors are mediated by a container runtime, QiMeng-PerceptOS remains capable of accurately aligning hardware-software fingerprints with the LLM's semantic space, demonstrating superior cross-environment generalization.

### E.5. Comparison with Semi-Automatic Methods

The experimental data in Table 13 demonstrates that QiMeng-PerceptOS consistently and significantly outperforms all semi-automated baselines across diverse workloads and LLM backends. Notably, the performance of AutoOS*—which optimizes each application directly rather than enhancing various OS components while preserving AutoOS's original search configurations—exhibits considerable instability, occasionally underperforming even the vanilla AutoOS. In contrast, QiMeng-PerceptOS substantially surpasses AutoOS*. This performance gap indicates that dynamic semantic alignment is more effective at unlocking kernel potential than merely altering a single variable of the optimization objective within the existing static framework.

To further contextualize the efficacy of QiMeng-PerceptOS, we conducted a targeted comparative study with BYOS, a state-of-the-art semi-automated system that incorporates expert-curated knowledge graphs. Given BYOS's reliance on proprietary components and manual expert intervention, we treat it as an additional high-level baseline to benchmark the precision limits of knowledge-driven yet non-perceptual frameworks.

As summarized in Table 13, the results reveal an intriguing "performance stratification": BYOS yields gains over the vanilla AutoOS only when utilizing GPT-4o-mini as the backbone. Surprisingly, when paired with DeepSeek-V3.2 or Gemini-3-pro-preview, BYOS delivers lower throughput improvements than the baseline AutoOS. This trend suggests a "semantic friction" between BYOS's static, expert-defined knowledge layer and the internal reasoning mechanisms of specific LLMs. When a model fails to effectively synthesize these rigid priors, static knowledge may act as structural noise rather than a guiding signal, thereby hindering exploration within the ultra-high-dimensional configuration space.

In contrast, QiMeng-PerceptOS maintains robust and leading performance across all evaluated LLM backbones. The superiority of QiMeng-PerceptOS over these semi-automated approaches is twofold: first, it achieves full end-to-end automation, eliminating the laborious expert intervention required by its counterparts; second, it overcomes the "Perception

*Table 13.* Detailed Comparative Study of QiMeng-PerceptOS against Semi-Automated Baselines. Relative throughput gains for 15 experimental permutations (5 workloads × 3 LLMs) compared to the default configuration. Values in the "Improv." row represent the absolute percentage point improvement of QiMeng-PerceptOS over BYOS and AutoOS*, respectively. This study evaluates the effectiveness of semantic-aware alignment in mitigating the reasoning bottlenecks inherent in static knowledge-based systems

| System Environment | LLM | Setting | Redis | Apache | AES | PostgreSQL | RAG |
|---|---|---|---|---|---|---|---|
| PC (Intel, Fedora 42) | GPT-4o-mini | AutoOS | 108.2% | 122.5% | 117.7% | 106.5% | 102.2% |
| | | BYOS | 122.7% | 105.4% | 110.0% | 110.7% | 125.7% |
| | | AutoOS* | 119.2% | 126.1% | 111.3% | 112.1% | 106.3% |
| | | QiMeng-PerceptOS | **147.0%** | **139.7%** | **116.8%** | **112.3%** | **133.8%** |
| | | *Improv.* | ↑24.3%/↑27.8% | ↑34.3%/↑13.6% | ↑6.8%/↑5.5% | ↑1.6%/↑0.2% | ↑8.1%/↑27.5% |
| | Deepseek-V3.2 | AutoOS | 121.6% | 137.2% | 111.9% | 107.2% | 109.7% |
| | | BYOS | 110.9% | 105.5% | 107.6% | 100.0% | 104.1% |
| | | AutoOS* | 121.3% | 137.4% | 114.7% | 107.6% | 110.2% |
| | | QiMeng-PerceptOS | **143.7%** | **147.9%** | **118.4%** | **116.0%** | **133.7%** |
| | | *Improv.* | ↑32.8%/↑22.4% | ↑42.4%/↑10.5% | ↑10.8%/↑3.7% | ↑16.0%/↑8.4% | ↑29.6%/↑23.5% |
| | Gemini-3-pro-preview | AutoOS | 123.7% | 149.1% | 121.2% | 113.9% | 116.5% |
| | | BYOS | 120.8% | 109.4% | 107.5% | 100.0% | 104.3% |
| | | AutoOS* | 122.1% | 141.7% | 113.9% | 122.5% | 129.1% |
| | | QiMeng-PerceptOS | **145.6%** | **165.7%** | **118.8%** | 115.5% | **135.5%** |
| | | *Improv.* | ↑24.8%/↑23.5% | ↑56.3%/↑24.0% | ↑11.3%/↑4.9% | ↑15.5%/↑-7.0% | ↑31.2%/↑6.4% |

Deficit" inherent in knowledge-driven frameworks. While manually crafted prompts remain "blind" to dynamic system behaviors—such as varying I/O patterns and hardware telemetry—QiMeng-PerceptOS bridges the gap between low-level system states and high-level reasoning via its closed-loop perception module. These findings establish that for optimizing complex, OS-intensive workloads, dynamic semantic alignment is fundamentally more effective and generalizable than reliance on static expert priors.

## E.6. Detailed Discussion on AutoOS's Cross-Platform Performance

As shown in Table 3, AutoOS exhibits limited relative performance gains on high-core-count workstation platforms when using its default configuration strategies. Our in-depth analysis attributes this to a fundamental mismatch between the framework's configuration heuristics and the underlying system architecture. AutoOS's default configuration strategies lack cross-platform perception, leading to configuration recommendations biased toward low-core-count environments (e.g., standard PCs).

For instance, when optimizing the RAG workload, AutoOS with DeepSeek-v3.2 enables CONFIG_SLUB_TINY and disables CONFIG_TRANSPARENT_HUGEPAGE. While these configurations reduce memory overhead—a beneficial trade-off on PC-class hardware—they trigger significant lock contention and TLB misses on high-speed, multi-core workstations.

To validate this, we conducted a cross-platform transplant experiment. We ported both the default workstation kernel (A) and the AutoOS with DeepSeek-v3.2-optimized kernel generated on workstation (B) to the PC (AMD, Ubuntu 22.04), keeping all other components identical. The results demonstrate that while kernel B yields negligible gains on the workstation, it produces substantial performance improvements compared to A on the PC: 185% for Redis, 171% for Apache, 140% for AES, 140% for PostgreSQL, and 114% for RAG. This confirms that AutoOS's configuration recommendations are intrinsically biased toward PC-class hardware. The stochasticity inherently present in AutoOS's search process remains insufficient to overcome its blindness to low-level hardware telemetry, further underscoring the necessity of the dynamic, perception-grounded optimization provided by QiMeng-PerceptOS.

## E.7. Trade-Off Study of Throughput and Latency

The trade-off between throughput and latency depends on specific scenario needs of users. QiMeng-PerceptOS inherently supports this flexibility. To demonstrate, we modified the optimization target of the GPT-4o-mini-based QiMeng-PerceptOS on PC Ubuntu to prioritize Redis latency rather than throughput.

Table 14 exhibits the flexibility of QiMeng-PerceptOS in addressing the inherent trade-offs between throughput and latency. Under the Latency-first objective, the system reduces p99 tail latency to 31.3% of the default; conversely, the Throughput-first objective drives a 2.78x improvement in request volume. Notably, both configurations significantly outperform the default configuration settings, demonstrating that users can flexibly switch the optimization goals of QiMeng-PerceptOS to perform trade-offs based on whether their specific use case prioritizes latency or throughput.

*Table 14.* An example to show how to handle trade-off between throughout and latency when optimizing OS for Redis via GPT-4o-mini-based QiMeng-PerceptOS on PC ubuntu.

| Optimization Target | Throughput (↑) | Latency (p99) (↓) |
|---|---|---|
| Default | 100% | 100% |
| Throughput-first | 278.5% | 35.2% |
| Latency-first | 250.5% | 31.3% |

## F. Detailed Experimental Results on the Ubuntu PC

In this section, we report the granular evaluation metrics and performance outcomes for five software applications on an Ubuntu PC, supplementing the aggregate percentage gains discussed in Sec. 4.

As summarized in Table 15, QiMeng-PerceptOS yields substantial throughput improvements across all Redis subcommands. For the SET command, QiMeng-PerceptOS achieves a throughput of 320.51 kRPS, a significant improvement over the vanilla's 105.26 kRPS. Similarly, the ZADD command reaches 318.47 kRPS under QiMeng-PerceptOS, compared to 106.61 kRPS for the vanilla setup. This consistent performance trajectory is also observed in the other four benchmarked applications, further validating the efficacy of the proposed system.

## G. Case Study of the Optimized Configuration

In this section, we present some representative options optimized by GPT-4o-mini-based QiMeng-PerceptOS for Apache on the Ubuntu PC in Table 16. Default operating system kernels prioritize general compatibility, which results in redundant checks and conservative strategies that introduce non-negligible overhead. QiMeng-PerceptOS employs a set of targeted optimization principles: disabling redundant overhead, amplifying bottleneck resources, and matching optimal algorithms. These strategies collectively reduce micro-level latency to achieve macro-level performance gains.

*Table 15.* Detailed performance optimization results across multiple applications on the Ubuntu PC. Average performance shows relative performance compared to the default configuration.

| Application | Test | Default Configuration | Vanilla | AutoOS | QiMeng-PerceptOS |
|---|---|---|---|---|---|
| Redis (kRPS) | PING_INLINE | 101.21 | 106.84 | 233.64 | 320.51 |
| | PING_MBULK | 102.46 | 104.82 | 231.48 | 318.47 |
| | SET | 100.60 | 105.26 | 232.56 | 320.51 |
| | GET | 101.63 | 105.93 | 231.48 | 318.47 |
| | INCR | 103.52 | 106.38 | 231.48 | 314.47 |
| | LPUSH | 103.31 | 106.84 | 232.56 | 316.46 |
| | RPUSH | 102.04 | 107.07 | 231.48 | 320.51 |
| | LPOP | 102.25 | 107.07 | 230.41 | 322.58 |
| | RPOP | 102.25 | 107.30 | 232.56 | 320.51 |
| | SADD | 102.25 | 106.84 | 230.41 | 318.47 |
| | HSET | 101.42 | 106.84 | 228.31 | 316.46 |
| | SPOP | 101.83 | 106.16 | 230.41 | 320.51 |
| | ZADD | 101.63 | 106.61 | 228.31 | 318.47 |
| | ZPOPMIN | 101.42 | 106.84 | 230.41 | 320.51 |
| | LRANGE_100 | 70.92 | 73.75 | 116.28 | 138.89 |
| | LRANGE_300 | 33.69 | 35.84 | 42.59 | 47.26 |
| | LRANGE_500 | 24.41 | 25.85 | 29.33 | 31.41 |
| | LRANGE_600 | 21.16 | 22.82 | 25.27 | 26.47 |
| | MSET | 103.09 | 107.99 | 210.08 | 260.42 |
| | XADD | 103.09 | 107.53 | 227.27 | 320.51 |
| | **Average Performance** | **100%** | **104.8%** | **207.3%** | **278.5%** |
| Apache (RPS) | Average Throughput | 16630.0 | 16855.4 | 30232.5 | 49205.8 |
| | **Average Performance** | **100%** | **101.4%** | **181.8%** | **295.9%** |
| AES (MiB/s) | **Encryption** | | | | |
| | cbc128b | 1293.2 | 1261.2 | 1438.1 | 1494.6 |
| | cbc256b | 994.8 | 998.5 | 1078.0 | 1110.1 |
| | xts256b | 3197.0 | 3265.9 | 4896.1 | 5436.5 |
| | xts512b | 2867.5 | 2886.2 | 4047.4 | 4404.1 |
| | **Decryption** | | | | |
| | cbc128b | 3420.2 | 3376.6 | 5850.9 | 6588.9 |
| | cbc256b | 3163.0 | 3203.6 | 4743.7 | 5253.7 |
| | xts256b | 3197.9 | 3251.0 | 4886.5 | 5446.6 |
| | xts512b | 2923.1 | 2933.9 | 4053.0 | 4447.3 |
| | **Average Performance** | **100%** | **101.1%** | **140.8%** | **154.0%** |
| PostgreSQL (TPS) | Average Throughput | 3942.8 | 3987.5 | 5628.8 | 6091.5 |
| | **Average Performance** | **100%** | **101.1%** | **142.8%** | **154.5%** |
| RAG (QPS) | Average Throughput | 6.35 | 6.48 | 7.20 | 7.89 |
| | **Average Performance** | **100%** | **102.1%** | **113.4%** | **124.3%** |

*Table 16.* The optimization result on the Ubuntu PC optimizing apache with GPT-4o-mini-based QiMeng-PerceptOS. In the list, a plus sign (+) indicates default disabled settings that are now enabled, while a minus sign (-) indicates the opposite change.

| Related call stacks | Category | Representative options | Modifications | Description of options |
|---|---|---|---|---|
| \_\_vdso\_gettimeofday | Timekeeping (4) | CONFIG\_COMPAT\_VDSO | + | Provide compat vDSO support for fast user space time reads. |
| | | CONFIG\_NO\_HZ\_IDLE | + | Enable tickless idle mode to reduce periodic timer interrupts. |
| | | CONFIG\_CLOCKSOURCE\_WATCHDOG\_MAX\_SKEW\_US | 100→1000 | Set the maximum allowed clocksource skew before reporting issues. |
| \_\_libc\_connect \_\_x64\_sys\_connect \_\_sys\_connect security\_socket\_connect current\_check\_access\_socket | Security (13) | CONFIG\_SECURITY | - | Disable Linux Security Module support and related security hooks. |
| | | CONFIG\_AUDIT | - | Disable kernel auditing for security and compliance logging. |
| | | CONFIG\_HARDENED\_USERCOPY | - | Disable hardened usercopy checks that detect invalid copy operations. |
| | | CONFIG\_SECURITY\_NETWORK\_XFRM | - | Disable security hooks for network transformation policy. |
| | | CONFIG\_SECURITY\_APPARMOR | - | Disable AppArmor support for path based access control. |
| inet\_stream\_connect tcp\_v4\_do\_rcv tcp\_rcv\_state\_process tcp\_send\_ack tcp\_sync\_mss | Transport (9) | CONFIG\_TLS | - | Disable kernel TLS support for offloading record handling in the kernel. |
| | | CONFIG\_TCP\_CONG\_LP | - | Disable TCP low priority congestion control. |
| | | CONFIG\_INET\_DIAG | - | Disable INET diagnostics interfaces for socket introspection. |
| | | CONFIG\_INET\_DIAG\_DESTROY | - | Disable privileged teardown of sockets via INET diagnostics. |
| | | CONFIG\_IP\_ROUTE\_VERBOSE | - | Disable verbose messages for routing related events. |
| ip\_output ip\_finish\_output2 \_\_dev\_queue\_xmit validate\_xmit\_skb \_\_napi\_poll | Packet processing (23) | CONFIG\_MAX\_SKB\_FRAGS | 17→45 | Set the maximum number of fragments per socket buffer. |
| | | CONFIG\_NET\_CLS\_MATCHALL | - | Disable matchall classifier support in traffic control. |
| | | CONFIG\_NET\_CLS\_CGROUP | - | Disable cgroup based classifier support in traffic control. |
| | | CONFIG\_NET\_ACT\_BPF | - | Disable BPF based actions for traffic control. |
| | | CONFIG\_NET\_IFE | - | Disable ingress frame encapsulation support. |
| nf\_hook\_slow nft\_do\_chain\_ipv4 nft\_do\_chain nft\_counter\_eval ip\_local\_deliver | Netfilter and nftables (9) | CONFIG\_NETFILTER | - | Disable the netfilter framework for packet filtering and manipulation. |
| | | CONFIG\_XFRM\_USER | - | Disable network transformation user space configuration interface. |
| | | CONFIG\_XFRM\_INTERFACE | - | Disable network transformation interfaces. |
| | | CONFIG\_XFRM\_STATISTICS | - | Disable collection of network transformation statistics. |
| | | CONFIG\_NET\_KEY | - | Disable key management socket interface for network policy. |
| \_\_vdso\_gettimeofday do\_syscall\_64 handle\_softirqs net\_rx\_action \_\_napi\_poll | CPU and power (14) | CONFIG\_CPU\_MITIGATIONS | - | Disable CPU vulnerability mitigations. |
| | | CONFIG\_SCHED\_AUTOGROUP | - | Disable scheduler automatic process group support. |
| | | CONFIG\_PROFILING | - | Disable profiling support. |
| | | CONFIG\_ENERGY\_MODEL | - | Disable energy model support for scheduling decisions. |
| | | CONFIG\_PM | - | Disable power management support. |
| \_\_memcpy \_\_memset process\_backlog \_\_netif\_receive\_skb ip\_rcv\_finish | Memory management (8) | CONFIG\_TRANSPARENT\_HUGEPAGE\_ALWAYS | + | Always use transparent huge pages. |
| | | CONFIG\_USERFAULTFD | - | Disable userfaultfd for user space page fault handling. |
| | | CONFIG\_IDLE\_PAGE\_TRACKING | - | Disable idle page tracking support. |
| | | CONFIG\_DAMON | + | Enable data access monitoring infrastructure. |
| ... | And others | CONFIG\_FTRACE | - | Disable function tracing support. |
| | | CONFIG\_SAMPLES | - | Disable kernel sample code. |
| | | CONFIG\_RUNTIME\_TESTING\_MENU | - | Disable runtime testing configuration options. |
| | | CONFIG\_MEMTEST | - | Disable built in memory test support. |
| | | CONFIG\_STRIP\_ASM\_SYMS | + | Strip assembler symbols to reduce kernel image size. |

