# OpenReview forum: "QiMeng-PerceptOS: Semantic-Aware Kernel Optimization for OS-Intensive Workloads via Hardware-Software Alignment"
_ICML.cc/2026/Conference — ICML 2026 regular_

### Official Review · Reviewer_xsCp · 2026-03-04

**Soundness:** 3
**Presentation:** 3
**Significance:** 3
**Originality:** 3
**Overall Recommendation:** 4
**Confidence:** 4

**Summary:**

This paper presents PerceptOS, an autonomous framework for optimizing application-specific OS kernels. The core insight is that existing LLM-based methods suffer from a static “semantic mismatch” between the LLM’s general reasoning and low-level system dynamics. To address this, PerceptOS shifts to a perception-driven, closed-loop tuning paradigm via three collaborative modules: 1) a Perception Module that distills raw hardware/software telemetry into high-fidelity semantic fingerprints; 2) a Global Search Module utilizing a Bi-level Hierarchical Induction Tree (BHIT) for efficient navigation and pruning; and 3) a Posterior Enhancement Module that suppresses hallucinations via historical trajectory synthesis. Experiments across diverse hardware, OS distributions, and applications (e.g., Redis, Apache, PostgreSQL) demonstrate its efficacy, achieving up to 296.6% of default Redis throughput and outperforming SOTA baselines.

**Compliance With Llm Reviewing Policy:**

Affirmed.

**Final Justification:**

I believe the work presented in the paper matches the final score I assigned in the review.

**Key Questions For Authors:**

1.  **Deeper Comparison with Knowledge-Graph Methods:** The paper compares against BYOS as a semi-automated baseline. A more profound question is: If PerceptOS were also augmented with a **carefully curated, expert-level OS knowledge base** similar to BYOS, would its performance see **further improvement**, or has the introduction of dynamic perception already sufficiently compensated for or even surpassed the need for static knowledge? This would help clarify the relative importance of “dynamic perception” versus “static knowledge” in this task.
2.  **Dependence on LLM Capability Assumptions:** Experiments show that even a lightweight model like GPT-4o-mini performs excellently within the PerceptOS framework. Does this suggest the framework’s success relies heavily on its **prompt engineering and pipeline design**, reducing dependence on the LLM’s deep domain knowledge? If so, does it also imply the method’s bottleneck may lie in template design rather than the LLM’s reasoning capability?

**Limitations:**

Yes.
The authors responsibly discuss the method’s limitations in Appendix A, including limited effectiveness for compute-intensive tasks with low OS dependencies and inherent multi-dimensional trade-offs. It is recommended to briefly mention these boundaries in the main text’s “Future Work” section to help readers better understand the method’s scope. Additionally, discussion on potential challenges for deployment in very large-scale clusters or real-time systems could be considered.

**Strengths And Weaknesses:**

- **Correctness:** The argument is solid. The problem is well-defined, the methodology is detailed, and the core hypotheses are validated through comprehensive experiments. Ablation studies convincingly demonstrate the necessity of each module. The experimental setup considers various hardware, OS, and LLM backends, and the results show statistical robustness and reproducibility.
- **Clarity:** The paper is well-structured and logically coherent. The algorithm description and experimental design are replicable. The extensive appendices and provided templates offer valuable details.
- **Significance:**
  - **Strengths:** The work tackles a real and challenging problem—automated OS kernel tuning in large-scale, high-cost environments. The proposed perception-driven paradigm offers a new perspective on leveraging LLMs for system optimization and is thought-provoking.
  - **Weaknesses:** As noted by the reviewer and acknowledged by the paper itself (Appendix A), the method is limited in its effectiveness for applications with “inherent multi-dimensional trade-offs,” primarily excelling in scenarios with a relatively singular optimization target and high OS dependency. This somewhat limits its generality and breadth of application. Although it achieves remarkable results on specific types of “OS-intensive” workloads, positioning its contribution as a universal, revolutionary paradigm for system optimization might be premature. Its practical significance leans more towards providing an efficient engineering solution for a specific problem domain rather than a broadly applicable theoretical breakthrough.
- **Originality:**
  - **Strengths:** The paper clearly identifies the “semantic mismatch” as the core bottleneck and systematically decomposes it into three concrete issues: the perception-alignment dilemma, configuration fragmentation, and hallucination-induced search drift. The proposed closed-loop framework of “Perception-Global Search-Posterior Enhancement” represents a substantial improvement over existing static, open-loop LLM approaches. The dynamic transformation of raw system telemetry into LLM-understandable semantic fingerprints is a key innovation, enabling a paradigm shift from “black-box search” to “perception-driven tuning.”
  - **Weaknesses:** The core contribution seems more like a clever **engineering framework innovation**. It does not propose a novel machine learning model or theory but rather employs the LLM as an agent possessing both a **static knowledge base (albeit non-expert level)** and **dynamic planning capabilities (iterating based on feedback)**, constructing a closed-loop system capable of autonomous perception and exploration. While effective, its novelty lies primarily in **how existing components are orchestrated** (LLMs, templates) to solve a domain-specific problem. Although the paper emphasizes its distinction from methods like RL, its essence remains an interactive, budget-constrained optimization process driven by an agent (LLM).

---

> ### Author Rebuttal · Authors · 2026-03-31
>
> Thank you for your reviews!
>
> ## Weakness
>
> > W1-Method is limited... multi-dimensional trade-off
>
> We clarify that paper does not position PerceptOS as a "universal revolutionary paradigm for system optimization," but rather as a paradigm shift in 'LLM for optimization'—advancing from no/numerical perception to semantic perception
>
> (a) Clarification on Limitations
>
> Limited gains in compute-heavy or resource-competing scenarios stem from inherent system/mathematical constraints, not the optimization method failure. Please see detailed discussions in Reviewer-zuPW-weak-1&3
>
> (b) Generalizability
>
> PerceptOS is specifically suitable for high-dimensional, high-cost, and high-risk optimization. OS kernel tuning is only a representative case.Our insights can be leveraged for other tasks facing similar challenges, such as directly optimizing database or other software&complex system's compile-time configuration to accelerate workloads that depend on it. **These domains share the same high-dimensional and high-risk characteristics where semantic mismatch remains a hurdle**
>
> > W2-engineer...remains an interactive, budget-constrained optimization
>
> (a) PerceptOS is **the first to propose deep semantic perception to solve semantic mismatch** in existing optimization process of LLM. Traditional LLM methods generate configurations based on heuristic reasoning, creating “semantic mismatch” between LLM reasoning and actual low-level system states. A naive method would be letting LLM compare numerical performance changes and configuration edits across iterations to help the optimization. However, this faces severe attribution difficulties: specially in hyper-dimensional spaces, performance change result from the collective impact of many options, making it nearly impossible to work. This still remains a **shallow perception** that fails to capture actual systemic changes
>
> This table show difference in perception from traditional methods:
> | Method | Semantic Perception | Issues in High-Dim/High-Risk/High-Cost Tasks like OS |
>  | :--- | :--- | :--- |
> |Cost Model| No (Numeric-only) | fails on sparse samples|
> |RL| No (Vector-only) | High interaction cost|
> |Monte Carlo| No (Numeric-only) | High interaction cost|
> |Bayesian| No (Numeric-only) | Fails at 15k+ dims; risk of boot-failure|
>  |LLM (AutoOS/BYOS)| No (Numeric-only) | Severe Semantic Mismatch|
>
> (b) We believe that both applications of ML and ML theoretical are key topics of interest in ICML and top venues[1][2][3]. **AutoOS is ICML2024**. ICML2026 explicitly lists Applications as one of the topics of interest. By addressing a complex problem with significant real-world utility, we believe our insights **in LLM for optimization** can offer valuable takeaways for similar tasks(see Weak1b) and has contribution on application track.
>
> [1] Alpha-SQL: Zero-Shot Text-to-SQL using Monte Carlo Tree Search ICML2025
>
> [2] Debiasing LLMs for Faithful TableQA via Front-Door Intervention NIPS2025
>
> [3] ROGA: Scaling Generalist Agents for Office Productivity Tasks via Tool Generation ICLR2026
>
> ## Questions
> > Q1-Knowledge-Graph Method
>
> They represent two distinct trajectories:
>
> ●BYOS (Static Knowledge): Relies on expert-curated knowledge cards. Under perfect rules, it essentially functions as a rule-inference system
>
> ●PerceptOS (Semantic-Aware)
>
> The perfect knowledge is useful. However, the bottleneck of the static route is the **impracticality** of find "perfect" knowledge for every real-world scenario —especially when doc.is missing or of low quality. Table 11 shows BYOS’s rough rules often cause "semantic friction"
> PerceptOS prioritizes semantic alignment to eliminate the need for manual expert intervention, ensuring superior automation and generalization. We believe that if "perfect" static knowledge were available, the two are complementary, not exclusive
>
> > Q2-LLM Capability
>
> We contend that optimization bottlenecks lie in both LLM capability and semantic mismatch -**whereas prior work has largely focused on the former while neglecting the latter**
>
> (a) Semantic alignment does not replace the LLM's inherent reasoning capability. As shown in Table 3, more powerful models (e.g., Gemini3-pro-preview) significantly outperform lightweight models (e.g., GPT-4o-mini) within the PerceptOS framework. Moreover, PerceptOS further amplifies the performance of Gemini3-pro-preview— demonstrating a synergetic effect
>
> (b) Semantic perception is not just specific prompt engineering. As shown in Appendix B, our templates are simple. Furthermore, the system is not sensitive to template-level details (detailed see Reviewer-G5oA-W5)
>
> > discussion-large-scale or real-time
>
> Yes,we'll move the limitations to 'future work'
>
> Large-Scale: When hardware in cluster are different, people can explore federated optimization to share insights across nodes while adapting to local hardware
>
> Real-Time: people can establish offline pre-optimized libraries for dynamic switching and study on secure fallback mechanisms

---

> > ### Author Rebuttal · Reviewer_xsCp · 2026-04-01
> >
> > Thank you for your responses and clarifications on the questions, which offer a clearer understanding of PerceptOS's contributions. Based on these explanations, I believe the work presented in the paper matches the score I assigned in the review.

---

> > > ### Author Response · Authors · 2026-04-01
> > >
> > > We are glad that the rebuttal has addressed your concerns!  We appreciate the time and effort you have dedicated to reviewing our work. We also deeply value your recognition of our rigorous methodology and experimental design, as well as your acknowledgement that this work  'enabling a paradigm shift from “black-box search” to “perception-driven tuning.” '. We will carefully incorporate your suggestions to refine the paper.

---

### Official Review · Reviewer_G5oA · 2026-03-09

**Soundness:** 3
**Presentation:** 2
**Significance:** 3
**Originality:** 2
**Overall Recommendation:** 4
**Confidence:** 2

**Summary:**

The paper describes a tool for OS optimization with LLMs. The optimization framework consists of three modules for perception, global search and posterior enhancement. The framework addresses semantic mismatch between the LLM and the OS and it outperforms previous frameworks within 15 iterations.

**Compliance With Llm Reviewing Policy:**

Affirmed.

**Final Justification:**

The authors provided a detailed and helpful rebuttal that addressed my concerns and significantly improved the clarity of the paper. In particular, the additional explanations regarding the optimization process and design choices made the approach easier to understand, even for readers less familiar with OS-level optimization.

While my confidence in some of the deeper technical aspects remains limited due to my background, I find the work interesting, well-presented, and potentially useful. I therefore maintain my weak accept recommendation.

**Key Questions For Authors:**

no specific questions

**Limitations:**

no, I would advise the authors to consider including some reflection on robustness

**Strengths And Weaknesses:**

Even though I have limited background in operating system optimization, the paper presents its ideas in a very accessible way. The manuscript is written in a clear and well-structured manner, which allows the reader to follow the argument smoothly without frequently having to reinterpret terminology or concepts. This clarity is a strength of the paper.

At the same time, for readers who are less familiar with OS-level optimization, a more detailed explanation of why the proposed approach leads to performance improvements, e.g., how certain kernel parameters influence the throughput. That would help to contextualize the impact of the work. \
If I understand correctly, the main contribution of this framework is the architecture of the iteration cycle between the perception and global search with the pruning step. I would have liked more focus and in-depth description of that and how this particularly differs from existing methodology. In Figure 2 the Posterior Enhancement Module contains a component labeled “filter”. However, the text does not explicitly describe this step or use the word elsewhere. It appears to correspond to the pruning of configurations, but clarifying this would improve the readability of the framework.

The reported throughput improvements appear very large to me (acknowledging that I do not have extensive prior experience with OS kernel optimization). Presenting the results primarily relative to the default configuration may exaggerate the perceived magnitude of the gains. It might therefore be helpful to report improvements relative to the strongest baseline (SOTA approaches) as the primary comparison, rather than the default configuration, to provide a clearer sense of the relative performance improvements.

I assume that the templates provided in the appendix correspond to the prompts used for the LLM components. I am curious whether the authors conducted any experiments with variations of these prompts to test the robustness of the system. \
Additionally, the paper critiques BYOS for relying on human priors. However, since the proposed approach relies on prompts designed by humans, it would be useful to discuss how this differs conceptually from incorporating human priors. Clarifying this distinction would strengthen the comparison to existing methodology.

Additionally, some parts of the text that need a formal rewriting (punctuation, grammar, consistency, etc.): \
Instead of referring to sections with §3.2, I would suggest using hyper references. \
“trajectories ,” \
“that identifies primary feature.” \
Consistency across references: ”(Lines 6–29)”, “(see Alg.. 1, L8–L9” \
“Alg. 1(L10) ,the” \
You do not number equations in section 2, but in section 3 you do. If not referenced later on, the equation numbering is not necessary, unify \
NUMA topologies, and ISA extensions should be explained, I do not know what those are \
In 3.3 t is not explained. I assume it stands for the iterations? But I am still confused because here the configuration is called theta and before it was T. \
“guarantees the ultimate bootability of the optimized kernel” no fullstop \
“targets:Redis” \
“The example of benchmark is provided in Table 12 of App.F.” \
“App.E.2” \
“variations(See App.E.2).” \
Section 4.2 listing: closing parenthesis are missing: “(1”

---

> ### Author Rebuttal · Authors · 2026-03-31
>
> Thank you for review comments!
>
> ## Weakness
>
> > W1-Why improvements
>
> (a) It is due to distinct optimization process in PerceptOS from semantic perception. For example, when optimizing apache: Perception Module first extracts specific call stacks, such as _x64_sys_socket → inet_stream_connect and hardware. Global search identifies network bottlenecks, pruning irrelevant branches while uncovering deep-seated options like congestion control. Finally, the posterior enhancement module, at performance plateaus, re-audits the alignment between network option settings and stacks. Existing open-loop methods bases on heuristic reasoning with semantic mismatches  (see Tables 3 and 11).
>
> (b) Case Study. We summarized the optimization result on the Ubuntu PC optimizing apache with GPT-4o-mini-based PerceptOS **in this [link](https://anonymous.4open.science/api/repo/picx-images-hosting-DEED/file/option.7i0ykgwpml.webp?v=5b8a1009)**. This shows the example of modified options, categories, corresponding call stacks, and their specific impact.
>
> > W2- In-depth description & how this differs
>
> The fundamental shift in PerceptOS is the transition from blind or purely numerical perception to **deep semantic-aware perception**
> Existing LLM methods (AutoOS, BYOS) generate configurations based on heuristic reasoning, creating a “semantic gap” between LLM reasoning and actual low-level system states. A naive method would be letting LLM compare numerical performance changes and configuration edits across iterations to optimize. However, this faces severe attribution difficulties: specially in hyper-dimensional spaces, performance change result from the collective impact of many options, making it nearly impossible to work. This still remains a “shallow” perception that fails to capture actual systemic changes
>
> PerceptOS first introduces **semantic-aware perception** to fundamentally resolve attribution challenges and semantic mismatch through:
>
> Perception Module: Aligns low-level telemetry with LLM’s reasoning.
>
> BHIT: Aligns the hyper-dimensional configuration space with LLM’s reasoning.
>
> Posterior Enhancement Module: Rectify possible semantic misalignments between low-level telemetry and hyper-dimensional configuration space.
>
> Close-loop: enables the LLM to capture semantic shifts in system states post-modification and optimize accordingly.
>
>  We believe PerceptOS differs significantly from prior methods: Please see more discussion in Reviewer-xsCP-weak-2.
>
> > W3- 'Filter' In Fig 2
>
> The "filter" in Fig 2 corresponds to the sentence in Line 224: "Configurations exhibiting semantic incongruence with F—indicative of possible hallucinations—are pruned"
>  Specifically, the Posterior Enhancement module traverses the modified options within the elite configuration set in their original order. It re-audits each option against semantic fingerprints to determine if it is performance-regressive; if so, the option is filtered (reverted). See more discussion in Reviewer-zuPW-weak-2. We commit to update the sentence from "prune" to "filter" to ensure consistency.
>
> > W4-Improvements relative to SOTA
>
> Thank you for the suggestion. Tables 3 and 11 (gray rows) already report gains relative to the SOTA.
> We use the default configuration as a universal baseline because all methods (PerceptOS, AutoOS, BYOS, and Vanilla) start from this same point. This allows readers to easily compare PerceptOS's relative performance against any baseline.
>
> > W5-Prompt robustness
>
> LLM’s language understanding makes PerceptOS robust to phrasing. We recruited two men (Alice and Bob) to rewrite prompts while preserving their original meaning for Apache optimization on PC Fedora (GPT-4o-mini). Functional formatting prompts remained constant. Variations are **in this [link](https://anonymous.4open.science/api/repo/picx-images-hosting-DEED/file/template.7p46fwiv27.webp?v=d432640e)**.
>
> | Prompts | Performance (Fedora) |
> | :--- | :---: |
> | Our| 139.7% |
> | Alice | 134.4% |
> | Bob| 139.6% |
>
> It shows: (1) PerceptOS is without complex prompt engineering (2) Phrasing has limited impact on performance due to the robustness of our semantic-aware mechanism.
>
> > W6-Difference from BYOS
>
> **After the method is designed**, for new applications, BYOS still necessitates manual document collection and Knowledge Graph updates by experts. Besides, Table 11 shows constructing high-quality prior knowledge is often impractical. When documentation is missing or substandard, it fails. In contrast, PerceptOS eliminates expert intervention via semantic-aware perception, ensuring superior generalization and the fully autonomous pipeline. See more in Reviewer-xsCP-Q1
>
> > W7-Punctuation, grammar...
>
> Thank you. We commit to correcting all in the final version to ensure overall consistency and compliance with formatting standards: Update all section and appendix references to clickable hyperlinks, Punctuation,Citation Consistency, Clarity t is iteration, Add brief explanations for NUMA and ISA and so on

---

> > ### Author Rebuttal · Reviewer_G5oA · 2026-04-03
> >
> > Thank you very much for the thoughtful explanations. I am afraid I am not sufficiently familiar with the topic to fully understand the details provided in your response to W1, and this is due to gaps in my own background rather than any lack of clarity in your answer.
> > Regarding the other points I raised, I found the rebuttal helpful and clarifying, and it addressed the concerns I had.

---

> > > ### Author Response · Authors · 2026-04-05
> > >
> > > We are glad the rebuttal addressed your concerns. Thank you for your kind suggestions. For W1, we take this very seriously and fully understand that knowledge systems vary across fields. To make the discussion more accessible, we would like to restate W1 in a more intuitive and straightforward way.
> > >
> > > **How Optimizing OS Kernel Parameters Impacts Application Performance**
> > >
> > > Standard OS kernels prioritize general compatibility, leading to redundant checks and conservative strategies that incur overhead. PerceptOS optimizes by: Disabling redundant overhead + Amplifying bottleneck resources + Matching optimal algorithms, reducing micro-level latency for macro-level gains.  The categories of options optimized by PerceptOS (powered by GPT-4o-mini) for Apahe on an Ubuntu PC are presented in the table below. In intuitive terms, PerceptOS's optimization strategies for these major categories are as follows:
> > >
> > > | Category | Why It Makes Things Faster |
> > > | :--- | :--- |
> > > | CPU & Power Management | Stops constant resource checking and frequent timer interrupts. |
> > > | Memory Allocation & Management | Reduces unnecessary memory shuffling and enables faster request paths. |
> > > | Debugging & Observability | Disables slow logging and debug prints to save CPU cycles. |
> > > | Cryptography & Security | Unloads unused encryption algorithms, saving memory, CPU, and boot time. |
> > > | Networking & Protocol Stack | Removes redundant checks and uses smarter data-sending strategies. |
> > > | Filesystems & I/O | Removes layer checks and prioritizes read/write requests for lower latency. |
> > > | Platform & Hardware Buses | Stops initializing or polling legacy/disconnected hardware. |
> > > | Kernel General & Interfaces | Minimizes overhead from monitoring, randomization, and sleep cycles. |
> > >
> > > An example for certain parameter : Disabling CONFIG_XFRM_STATISTIC removes per-packet stats, reducing CPU instructions per packet for Apache, improving performance.
> > >
> > > **Fully Understanding Between Kernel Parameters and Application Performance is Difficult even for Experts:**
> > >
> > > ●Massive Parameter Space: Linux possesses over 15,000 kernel parameters, and this number continues to grow;
> > >
> > > ●Non-linear Interactions: Complex mutual influences exist among parameters;
> > >
> > > ●Environmental Sensitivity: Parameter behavior varies significantly across different hardware, OS versions, and workloads;
> > >
> > > ●Vast Differences in Application Characteristics: The optimization directions for Redis, databases, and Web servers are fundamentally distinct.
> > >
> > > This precisely explains why: (1) existing static and open-loop automated tuning methods struggle with semantic mismatch; and (2) even experienced developers struggle to achieve high-quality manual tuning.
> > >
> > > This is the exact value of PerceptOS: **Users can discover the optimal configuration without requiring expertise**.
> > >
> > > **Why Does PerceptOS Outperform Existing Static, Open-Loop LLM Optimization Methods?**
> > >
> > > Existing methods suffer from Semantic Mismatch including runtime blindness, configuration fragmentation, and search drift. PerceptOS introduces a semantic-aware optimization paradigm to solve them as previously said: PerceptOS can abstract semantic fingerprint, enable effective optimization in high-dimensional spaces on BHIT , capture actual system changes in optimization by semantic close-loop, while ensuring semantic alignment. In contrast, other methods can only rely on predefined templates and heuristic search, blindly guessing whether an option needs optimization.
> > >
> > > Thank you again for your effort and patience.

---

### Official Review · Reviewer_bEJG · 2026-03-13

**Soundness:** 3
**Presentation:** 3
**Significance:** 3
**Originality:** 2
**Overall Recommendation:** 4
**Confidence:** 3

**Summary:**

This paper proposes PerceptOS, a perception-driven framework for automated OS kernel optimization targeting application-specific workloads. Existing LLM-based methods suffer from semantic mismatch with low-level telemetry, leading to runtime blindness and search drift. PerceptOS addresses this through three modules: (1) Perception Module—distilling hardware/software telemetry into semantic fingerprints; (2) Global Search Module—BHIT (Bi-level Hierarchical Induction Tree) for hierarchical pruning and configuration search; (3) Posterior Enhancement Module—trajectory synthesis to suppress hallucinations. Experiments on Redis, Apache, AES, PostgreSQL, and RAG across three hardware/OS environments show PerceptOS achieves up to 296.6% of default Redis throughput and surpasses AutoOS by 32.6% within 15 iterations.

**Compliance With Llm Reviewing Policy:**

Affirmed.

**Final Justification:**

I keep my weak accept recommendation because the paper offers a clear and practically meaningful systems contribution with broad experiments and reasonable soundness, although the novelty is more in system integration than in fundamentally new methodology and some gains are uneven across settings. The rebuttal addressed my main concerns and strengthened the experimental justification, but it did not materially change my overall evaluation.

**Key Questions For Authors:**

Q1. **Posterior Enhancement necessity:** Table 6 shows no improvement for Apache and RAG with Posterior Enhancement. Could the authors explain (a) why this module fails to add value on these workloads, and (b) whether the module could be conditionally applied (e.g., only when search drift is detected)? *How this would change the review:* A clear rationale or conditional design would strengthen the Soundness rating.

Q2. **Workstation baseline and Table 8 consistency:** (a) On the Workstation, AutoOS achieves ~100% on several benchmarks. The paper attributes this to semantic mismatch when migrating static prompts to new platforms—could the authors add a brief note on whether AutoOS was re-run or adapted for Workstation? (b) Table 8 (App. E.1) reports AutoOS at 47.9% for Apache on Fedora at 15/24 iterations, while Table 3 reports 122.5% for the same setting. Could the authors verify if 47.9% is a typo (e.g., 122.5% or 147.9%)? *How this would change the review:* Clarification would address potential unfair comparison or data consistency concerns.

Q3. **Perception Module design:** The diversity-aware windowing (v=20, k=50) and LLM-based semantic discriminator for stack selection—how sensitive is performance to these hyperparameters? Is there ablation on v, k, or the number of representative stacks? *How this would change the review:* Ablation would strengthen the design justification.

**Limitations:**

yes

**Strengths And Weaknesses:**

**Strengths:**

S1. **Clear problem framing:** The paper identifies three concrete obstacles (perception blindness, context fragmentation, hallucination-induced drift) in LLM-based kernel tuning and motivates the closed-loop perception paradigm.

S2. **Comprehensive experimental setup:** Evaluation spans 5 workloads, 3 hardware platforms (AMD/Intel PCs, 24-core workstation), 3 OS distributions, and 3 LLMs. Table 3 provides a thorough comparison against Vanilla LLM and AutoOS.

S3. **Ablation validates module contributions:** Table 5 shows that removing HW fingerprint, SW telemetry, hierarchical summary, or closed-loop each degrades performance (e.g., 128.9% → 114.34% without HW). Table 6 demonstrates Posterior Enhancement adds marginal gains on Redis, AES, PostgreSQL.

S4. **Practical relevance:** The 40–70 min evaluation cost per iteration is well-motivated; PerceptOS achieves substantial gains within 15 iterations, which is practically meaningful.

**Weaknesses:**

W1. **Posterior Enhancement effect is limited** (Soundness): Table 6 shows that Posterior Enhancement yields no improvement on Apache (295.89% unchanged) and RAG (124.25% unchanged). Gains on Redis/AES/PostgreSQL are modest (e.g., Redis 260.95% → 278.53%). The claimed "suppress hallucinations via trajectory synthesis" lacks quantitative evidence; the module's necessity and design rationale need stronger justification.

W2. **Inconsistent gains across environments** (Soundness): On Fedora Intel PC, PerceptOS occasionally underperforms AutoOS (AES: -0.9%, PostgreSQL: -2.4% with GPT-4o-mini; AES: -2.4% with Gemini). On the Workstation, AutoOS often achieves ~100% (near default), suggesting possible baseline configuration or evaluation setup issues that may inflate PerceptOS's relative advantage.

W3. **Innovation is primarily engineering integration** (Originality): The three modules combine existing ideas—template-based prompting for telemetry compression, hierarchical tree search with caching, and posterior synthesis from elite memory. The BHIT structure and diversity-aware stack sampling are incremental rather than fundamentally novel. The "first closed-loop perception framework" claim is valid but the technical novelty of each component is modest.

W4. **Reproducibility:** No source code is provided. Appendix B offers detailed templates (B.1–B.8) with prompt structures and examples, but the multi-round orchestration, error handling, and end-to-end pipeline are not fully specified for reproduction.

---

> ### Author Rebuttal · Authors · 2026-03-31
>
> Thank you for review comments!
>
> ## Weakness
>
> > W1-Posterior Enhancement effect
>
> (a) no gains in some cases is normal: Posterior Enhancement aims to **recover** performance losses from hallucinations in the global search stage, not force gains. If the search phase already finds an optimal, hallucination-free config (e.g.Apache/RAG), performance remains unchanged with no doubt
>
>  (b) Necessity: At plateaus in the close-loop, extra global search often introduces additional bad options. Posterior Enhancement(PE) re-audits LLM-modified options in elite sets via semantic fingerprints to have a chance to break plateaus. We replaced "13 search+2 PE" with "13+2=15search" in table 6:
>
> | App | 13 Search | 15 Search| **13+2 PE** |
> | :--- | :---: | :---: | :---: |
> | Redis | 260.9% | 260.9% | **278.5%** |
> | Apache | 295.9% | 295.9% | 295.9%|
> | AES | 151.9% | 152.2% | **154.0%** |
> | Postgresql | 149.9% | 149.9% | **154.5%** |
> | RAG | 124.3% | 124.3% | 124.3% |
>
> When extra global searches failed, posterior enhancement achieved a 17.58% marginal breakthrough in Redis,confirming its value.  We report posterior enhancement module has a 35% statistical probability of gain across all tests -3hardware & 5applications & 3LLM
>
> > W2-Inconsistent gains across environments
>
> Variations in improvement percentages are normal in system optimization, as final performance is jointly determined by hardware, kernel, and OS . Fedora/Intel PCs and Ubuntu workstations differ significantly in hardware resources and OS version; their distinct base performance levels naturally lead to different improvement percentages.
>
> AutoOS’s static prompts may suit specific scenarios,but its generalization is limited by the semantic mismatch. In table 3, PerceptOS significantly outperforms AutoOS in the vast majority of cases (e.g., Redis+32.6% on Fedora/Intel via GPT-4o-mini compared to AutoOS), with only minor fluctuations in rare instances. To ensure fairness: We used original AutoOS and BYOS repositories. All tests used identical random seeds. OS were obtained from official distributors.  Evaluation remained strictly consistent across all baselines.
>
> > W3-innovation and component
>
> We believe good innovation is **simple and effective**. Our core innovation of  PerceptOS is the shift from "no-aware" or "numerical-aware" to **'semantic perception'** in 'LLM for optimation'. Three components are just to realize this shift. Each also with a distinct departure from prior works:
>
> Perception: First to align low-level telemetry (e.g., system call stacks) with LLM's semantic space—beyond numerical indicators like IPC.
>
> Search: First to propose BHIT dual-layer induction, aligning high-dimensional configuration space with low-level telemetry.
>
> Posterior Enhancement: First to audit historical configurations via semantic fingerprints, recovering semantic mismatches in LLM.
>
> Due to space limit. please see details in Reviewer-xsCP-weak-2
>
>
> > W4-Reproducibility
>
> Implementation details are provided in the paper: Pseudo-code (Alg. 1); Multi-round orchestration & pipeline (Fig. 2 & Sec. 3); Error handling (Sec. 3.4, L229-243).
>
> We **commit to fully open-sourcing** the code, experimental setups upon publication to serve as a strong baseline for future research. Besides, Table 9 shows that PerceptOS achieves high reproducibility across different random seeds
>
> ## Questions:
>
> > Q1-(a) why not to add value (b) Conditional Application
>
> (a) See weak1 (b)We do not use a "detect drift before triggering" approach because drift itself is difficult to identify directly (high-dimensional attribution is hard). Instead, this module serves as **trial-and-error detection**, after the closed-loop search reaches a performance plateau. It re-audits elite configurations via semantic fingerprints: if performance improves after filtering misaligned options, drift/hallucination may exist
>
> > Q2-(a) Workstation baseline (b)Table 8:
>
> (a) No re-tuning. See weak 2.
>
> (b) Thank you for pointing this out. This was a data entry error; the 47.9% in Table 8 should be 122.5%, consistent with Table 3. We apologize and will correct this in the final version.
>
> > Q3-Perception hyperparameters
>
> We conduct an ablation study for this, optimizing Apache on a Ubuntu workstation using GPT-4o-mini:
>
> | Setting| v=0, k=0 | v=0, k=50 | v=20, k=50 | v=40, k=80 |
>  | :--- | :---: | :---: | :---: | :---: |
> | Performance| 116.4% | 124.6% | 128.9% | 127.5% |
>
> It shows:(a) Discriminator as a Safety Net: Moving from $k=0$ to $k=50$ (116.4% to 124.6%) with $v=0$ proves the LLM-based discriminator prevents losing essential call stacks when v is small.(b) Stability via Fixed Windows: $v=20, k=50$ outperforming $v=0, k=50$ shows that fixed top-$v$ stacks ensure core features are not omitted despite LLM instability. (c) Performance parity between $v=40, k=80$ and $v=20, k=50$ indicates performance is insensitive to moderate parameter increases for semantic fingerprints effectively shield the LLM from redundant information.

---

> > ### Author Rebuttal · Reviewer_bEJG · 2026-04-03
> >
> > The rebuttal resolves some of my concerns, but not all of them fully.
> >
> > I appreciate the clarification that the 47.9% value in Table 8 was a typo, and the added ablation on the Perception Module hyperparameters is helpful. The additional comparison between 13-search, 15-search, and 13+2 PE also makes the role of Posterior Enhancement clearer.
> >
> > That said, I still have two follow-up questions. First, the usefulness of Posterior Enhancement still appears workload-dependent. Could the authors report the claimed “35% probability of gain” more precisely in the final version, including the exact evaluation set and how this statistic is computed? Second, regarding the workstation results where AutoOS stays close to default on several tasks, could the authors add a clearer explanation in the paper of why this happens and whether any part of the result is sensitive to the particular prompt/template design of AutoOS?
> >
> > Overall, the rebuttal improves the paper, but my remaining concerns are only partially resolved, so I keep my score unchanged.

---

> > > ### Author Response · Authors · 2026-04-05
> > >
> > > > **Q1-Posterior Enhancement(PE)**
> > >
> > > The results and evaluation set are from 45 scenarios in table 3 (3 LLMs × 5 workloads × 3 hardware)
> > >
> > > The table below shows the **16 cases** (out of 45) where PE produced measurable gains (>0%), yielding a success probability of 16/45 = 35.6% in this set, while others likely reached optima in global search:
> > >
> > > |Application|Hardware|LLM|Gain|
> > > |-|-|-|-|
> > > |**Redis** |Ubuntu PC| 4o-mini |**+17.6%**|
> > > ||Fedora PC|DeepSeek|**+1.5%**|
> > > ||Fedora PC|Gemini|+0.6%|
> > > ||Workstation|4o-mini|**+3.8%**|
> > > ||Workstation|DeepSeek|**+3.7%**|
> > > |**AES**|Ubuntu PC|4o-mini|**+2.1%**|
> > > ||Ubuntu PC|DeepSeek|**+1.4%**|
> > > ||Workstation|Gemini|+0.7%|
> > > |**PostgreSQL**|Ubuntu PC|4o-mini|**+4.6%**|
> > > ||Fedora PC|4o-mini|**+6.0%**|
> > > ||Fedora PC|DeepSeek|**+8.3%**|
> > > ||Fedora PC|Gemini|**+6.3%**|
> > > ||Workstation|4o-mini|**+3.5%**|
> > > |**RAG**|Fedora PC| 4o-mini |+0.2%|
> > > || Workstation |DeepSeek|**+2.7%**|
> > > || Workstation|Gemini|**+1.5%**|
> > >
> > > It shows PE gains across Redis, AES, PostgreSQL, RAG, with Apache being the sole exception in this set.  However, this does not indicate workload dependence in PE. Two additional sources clarify: First, in the experiment in Table 10 (optimizing Apache in docker via GPT-4o-mini on workstation) , PE successfully show gains and recover the performance.  Second, to investigate this problem deeply, we conducted experiments on PC AMD(Ubuntu) using O1-based PerceptOS to optimize Apache. PE in this scenario again yielded an improvement. The results are below:
> > >
> > > **Results on Apache in more senarios**
> > >
> > > |PerceptOS Setting|13 Search|13+2PE|Gain|
> > > |-|-|-|-|
> > > |Workstation + Docker (4o-mini) |118.0%| **120.6%** |+2.6%|
> > > |Ubuntu PC (O1) |309.5%|**315.2%**|+5.7%|
> > >
> > > It shows PE  not exclusively tied to any particular workload. Though the probability of gain may vary to some extent across workloads, it does not contradict the design philosophy of PE as a lightweight trial‑and‑error exploration and our contribution of transforming the LLM optimization paradigm into  semantic perception.
> > >
> > > > **Q2-Detailed Explanation**
> > >
> > > We used original prompts in AutoOS repository. Our in-depth study reveals that this is a classic case of mismatch between system architecture and configuration strategies.
> > >
> > > ***Problems in AutoOS Template***
> > >
> > > AutoOS exhibits two characteristics:
> > >
> > > 1.Its prompts lack perception mechanisms, failing to adapt to underlying hardware
> > >
> > > 2.It introduces a degree of stochasticity during pruning and configuration recommendation to explore optimal settings for every cases
> > >
> > > However, the design of this present inherent limitations—randomness can assist exploration but can't compensate for a "blindness" to low-level system behavior. We find that AutoOS's configurations still lean toward optimizations suited for machines with low CPU core counts like PC.
> > >
> > > For example, the impaction for Rag of DeepSeek-based AutoOS modifications on workstation in table3 is different across different hardware. (options available at **[link](https://anonymous.4open.science/api/repo/picx-images-hosting-DEED/file/option1.92qprfnexs.webp?v=b90c2b33)**. Its strategy is "minimalist, low overhead." Here are 2 examples:
> > >
> > > (a) CONFIG_SLUB_TINY=y
> > >
> > > ●PC good (4–8 cores): Reduced memory overhead;may by beneficial for lower overhead
> > >
> > > ●Workstation Fail: On 64+ cores, collapsing 64 "toll booths" into one triggers massive lock contention, tanking throughput
> > >
> > > (b) CONFIG_TRANSPARENT_HUGEPAGE=n
> > >
> > > ●PC good : With lower bus bandwidth, 4KB page latency is negligible. This trades a minor performance hit for reduced memory overhead
> > >
> > > ●Workstation Fail: High-speed buses demand high throughput. 4KB pages cause constant TLB misses during large-scale vector searches, starving bandwidth
> > >
> > > PCs bottleneck on single-core efficiency; streamlining helps. But workstations shift to lock contention and bandwidth. That's why whis configuration doesn't work.
> > >
> > > ***Further Experiment***
> > >
> > > We performed a cross-platform test by porting both the default workstation kernel (A) and AutoOS(deepseek)-optimized kernel on workstation(B) to the PC (AMD, Ubuntu22.04). By keeping anything constant and only swapping kernels, we measured the resulting performance gains of B over A :
> > >
> > > | setting |performance|
> > > |-|-|
> > > |Redis|185%|
> > > |Apache|171%|
> > > |Aes|140%|
> > > |PostgreSQL|140%|
> > > |Rag|114%|
> > >
> > > Despite nearly no gain on workstation, transferring the AutoOS-optimized kernel to PC (AMD, Ubuntu 22.04) scenario still improved performance across 5 applications. This indicates that AutoOS running on workstation often generates configurations biased to machine with low CPU cores, despite introducing some randomness, explaining limited workstation gains and semantic mismatch.
> > >
> > > > **Commitment**
> > >
> > > Thank you very much for your constructive reviews. We commit to include in the final version carefully: (1) the 15-vs.13+2 PE comparison，"35%"  and evaluation set details  (2) analysis of AutoOS's  performance on workstation. (3) ablation of 'v, k' hyperparameters
> > >
> > > We'll open-source the code and fix typos.

---

### Official Review · Reviewer_zuPW · 2026-03-21

**Soundness:** 3
**Presentation:** 2
**Significance:** 2
**Originality:** 2
**Overall Recommendation:** 3
**Confidence:** 4

**Summary:**

PerceptOS is a closed-loop, perception-driven framework that automatically tunes Linux kernel compile-time configurations to boost throughput for OS-intensive applications. The configuration space is massive, and each evaluation takes 40 to 70 minutes, with a real risk of boot failure. The paper identifies semantic mismatch as the central barrier to LLM-based kernel tuning, in which high-level reasoning does not align with low-level runtime behavior. PerceptOS addresses this with a perception module that compresses hardware descriptors and software telemetry into semantic fingerprints, a hierarchical search structure (BHIT) that navigates the config space. Evals are strong.

**Compliance With Llm Reviewing Policy:**

Affirmed.

**Final Justification:**

Dear authors, I acknowledge reading the rebuttal. Thank you for putting in the effort.

I agree that you have clarified the fairness of the comparison (iteration budgets) and provided some evidence of robustness across seeds and hardware, which I appreciate. However, my main concerns remain only partially addressed and would require substantial new analysis to resolve:

– The posterior enhancement/hallucination suppression module is still supported mostly by anecdotal examples rather than a systematic ablation across benchmarks.

– The boundary of applicability (OS‑intensive vs compute‑intensive workloads) is not yet characterized in a way that lets practitioners know when PerceptOS is likely to help.

– Multi‑dimensional trade‑offs (throughput vs latency or cross‑workload effects) are largely unexplored empirically and very important to understand how to operate PerceptOS.

– The ML contribution remains primarily an engineering architecture built around an LLM rather than a clearly articulated new learning method or well‑formalized semantic alignment framework.

These points touch the core claims of the work and cannot be fully addressed in a short rebuttal; they would require a more substantial revision. Accordingly, I am keeping my overall recommendation as a weak reject.

**Key Questions For Authors:**

Thanks for the nice and interesting piece of work. I have a couple of questions for you.

* What is the machine learning contribution beyond using an LLM as a heuristic controller? The paper argues BO and RL are ineffective due to cost and fragility, but I would like to understand whether PerceptOS provides an ML-grounded alternative or is primarily an engineering architecture around an LLM. Clarifying this would help me assess the fit for ICML.

* How robust are the results across seeds, runs, and hardware?

* Can you please clarify whether the comparison to baselines is fair in terms of identical iteration budgets?

* If the key novelty is perception-driven closed-loop semantic alignment, what are the measurable ML takeaways that generalize beyond kernel tuning? I would like to understand whether this transfers to other high-cost, high-risk optimization tasks or whether the contribution is specific to this domain. I come from a systems background, and we really like to understand these nuances.

**Limitations:**

Yes.

**Strengths And Weaknesses:**

+ I like the semantic fingerprinting approach. Compressing hardware descriptors and software telemetry (hot call stacks, NUMA topology cues) into concise fingerprints is a practical way to anchor LLM reasoning without saturating the context window.

+ The BHIT is clever. Navigating a 15,000+ parameter space with hierarchical pruning and lazy induction caching is a solid engineering contribution. It makes the search tractable.

+ The results are strong.

- The authors acknowledge reduced impact for compute-intensive workloads with low OS dependency. This is honest, but it limits the scope of the contribution. I would like to understand better where the boundary is.

- Hallucination suppression via posterior enhancement is claimed, but I am not fully convinced it is well validated. Can you show me more directly that this component is doing what you say it does?

- Evaluation focuses on throughput. The paper acknowledges multi-dimensional trade-offs but does not explore them deeply. I come from a systems background, and we really like to understand these nuances.

---

> ### Author Rebuttal · Authors · 2026-03-31
>
> Thank you for review comments!
>
> ## Weakness
>
> >  W1-compute-intensive&contribution scope
>
> Tighter correlation between application bottlenecks and the OS yields higher OS optimization returns. Notably, the number of such applications is vast, such as Memcached, MySQL, Kafka, RabbitMQ, Apache, Nginx, Aes, Naive RAG ... As discussed in App. A, the limited impact on compute-intensive workloads like LLM training is because the bottleneck has little to do with OS, rather than optimization itself failing
>
> OS is only a representative case; PerceptOS's philosophy can be transferred to similar tasks. Please see the scope of the contribution in Question1
>
> > W2-How is hallucination suppression doing
>
> We clarify the design and hallucination mitigation:
>
> (a) Nature of Posterior Enhancement: Trial-and-error detection, not deterministic suppression
>
> Because directly identifying hallucinations in high-dimensional spaces is difficult , this module is an ex-post audit triggered at performance plateaus in the close-loop search. It re-audits **previously modified options by LLM** in elite configurations based on current semantic fingerprints:
>
> Improved performance after filtering indicates possible semantic drift/hallucinations (because current LLM logic differs from its prior decision)
>
> Unchanged performance suggests near-optimal semantic alignment
>
> (b) Case Study
>
> The module tries to recover from 2 typical hallucination types.  For example ,in global search stage, the final AES configuration In Table 6 modified 331 options compared to the default. The second audit of Posterior Enhancement Module attempted to filter 34 of these options, leading to further performance gains, including:
>
> ●Obvious Contradictions: CONFIG_BUG=y was introduced (in iteration 4) via flawed reasoning ("debug helps identify issues"), which contradicts the semantic fingerprint and the target. In the posterior enhancement phase, the LLM correctly identified its overhead and filtered it
>
> ●High-Entropy Oscillations: the history of CONFIG_UCLAMP_TASK in the elite set was $n \rightarrow y \rightarrow n$, indicating unstable LLM decision-making for it. The LLM re-proposed that $n$ would degrade performance, restoring it to $y$ for test
>
> Please see more in Reviewer-bEJG-weak-1
>
> > W3-multi-dimensional trade-offs
>
> We clarify that this trade-offs stem from the fact that optimizing OS for a specific application may lead to performance degradation in others, as different workloads have distinct kernel requirements. PerceptOS prioritizes single-application optimization, a strategy of immense practical value as core services (e.g., Apache ) typically run on dedicated production nodes for years in production.
>
> While multi-app aggregate metrics are deferred to future work, PerceptOS's architecture is inherently extensible for multi-objective expansion. By semantic alignment, the framework provides a grounded basis for navigating complex, non-linear trade-offs. This alignment further enables BHIT to effectively prune configuration spaces under multi-dimensional constraints.
>
> ## Questions
>
> > Q1-ML contribution & ICML Fit
>
> (a)PerceptOS can be seen an ML-grounded alternative. OS kernel optimization can be formalized as a classic ML problem with high dimensionality, expensive cost, fragility and non-linear dependencies between options. Traditional methods fails as Table 7 shows. Current LLM-based methods suffer from a severe semantic mismatch. PerceptOS provides an insight - introduce **semantic-aware perception** to align high level reasoning with low level semantic features. We believe this paradigm shift in "LLM for Optimization" is a meaningful contribution applicable to other similar problems. For example, the task of directly optimizing database or other software & complex system compile-time configuration to accelerate workloads that depend on it—**presents challenges similar** to those found in OS kernel optimization
>
> (b) ICML explicitly calls papers for Applications; Notably, the baseline **AutoOS is from ICML 2024**. PerceptOS resolves the critical semantic misalignment in such methods,  achieves significant gains and solves a difficult problem with strong realistic value (e.g., Redis +296%, needing only 34% of machines)
>
> See more discussion in Reviewer-xsCP-weak-2
>
>
> > Q2-Robustness
>
> PerceptOS shows robustness due to semantic close-loop. Table 3 shows it significantly outperforms SOTA across hardware , OSs, and applications.  Table 9 shows that shifting initial seeds by 10 and 20 across 5 applications yields stable results with negligible variance
>
> > Q3-Fair comparison
>
> It's fair: one iteration denotes one configuration tree traversal and one performance test for both AutoOS and PerceptOS. The performance test  is the primary bottleneck. Besides, Table 8 (extending 15->24) confirms that 15 iterations are sufficient to fairly demonstrate stable performance ceilings for all methods
>
> > Q4-ML takeaways
>
> Please see Q1 above & more discussions in Reviewer-xsCP-weak-2

---

> > ### Author Rebuttal · Reviewer_zuPW · 2026-04-05
> >
> > Dear authors, I acknowledge reading the rebuttal. Thank you for putting in the effort.
> >
> > I agree that you have clarified the fairness of the comparison (iteration budgets) and provided some evidence of robustness across seeds and hardware, which I appreciate. However, my main concerns remain only partially addressed and would require substantial new analysis to resolve:
> >
> > – The posterior enhancement/hallucination suppression module is still supported mostly by anecdotal examples rather than a systematic ablation across benchmarks.
> >
> > – The boundary of applicability (OS‑intensive vs compute‑intensive workloads) is not yet characterized in a way that lets practitioners know when PerceptOS is likely to help.
> >
> > – Multi‑dimensional trade‑offs (throughput vs latency or cross‑workload effects) are largely unexplored empirically and very important to understand how to operate PerceptOS.
> >
> > – The ML contribution remains primarily an engineering architecture built around an LLM rather than a clearly articulated new learning method or well‑formalized semantic alignment framework.
> >
> > These points touch the core claims of the work and cannot be fully addressed in a short rebuttal; they would require a more substantial revision. Accordingly, I am keeping my overall recommendation as a weak reject.

---

> > > ### Author Response · Authors · 2026-04-07
> > >
> > > Thank you for your effort and review. We note that some concerns were already addressed in our paper and  we can clarify the remaining concerns
> > >
> > > > Q1-Posterior Enhancement(PE)
> > >
> > > (a) In Reviewer bEJG-W1, we expanded Table 6 ("15 search" vs. "13+2 PE"), demonstrating PE's value when global search bottlenecks.  (b) **we have responsed to Reviewer bEJG-Q1 in second round,  a systematic ablation result across all 45 scenarios** (derived from Table 3). PE recovers performance in 35%(16/45) of cases in the set. Importantly, this is not a new experiment but a report of existing data from rounds 13–15 in table 3. (c)  Due to word limits, we apologize very much for being unable to include the data here and kindly direct you to response to Reviewer bEJG for details
> > >
> > > > Q2-Boundary
> > >
> > > (a) **Table 2 and L261 in paper has categorized comprehensive and classic interaction patterns with OS** in OS-intensive workloads, offering a broad framework for readers to identify when PerceptOS is likely to help. (b) Our response to W1 **lists further examples** of OS-intensive workloads ;Generally,it includes applications like network services, databases, and memory management which number is increasing. (c) Crucially, **App. A** demonstrates that GPU-intensive tasks ( LLM training) also see no gains from vanilla or AutoOS and helps readers identify effective scenarios for both PerceptOS and previous works
> > >
> > > > Q3-Trade-off
> > >
> > > (a) The trade-off between throughput and latency depends on specific scenario needs. Perceptos inherently supports this flexibility. To demonstrate, we modified the optimization target of the GPT-4o-mini-based PerceptOS on PC Ubuntu to prioritize Redis latency. The results is:
> > >
> > > |Redis|Throughput↑|Latency(p99)↓|
> > > |-|-|-|
> > > |Default|100%|100%|
> > > |Throughput-first|278.5%|35.2%|
> > > |Latency-first|250.5%|31.3%|
> > >
> > > It shows that users can flexibly switch the optimization goals of PerceptOS to perform trade-offs based on whether their specific use case prioritizes latency or throughput. We will include this discussion in the Appendix.
> > >
> > > (b) Cross-apps: At Line45、Sec 2 and W1, we explicitly stated that PerceptOS focuses on optimizing OS for a single application in certain scenario, for single service (e.g., Apache) typically run on dedicated nodes for years. As said in W3, how to define aggregate metrics for multi-app joint optimization to balance cross-workload effects is a matter for future research. This doesn't undermine our contribution: establishing the semantic-perception paradigm for LLM optimization
> > >
> > > > Q4-Novelty
> > >
> > > **Well Formalization**: We abstract OS tuning in Line46 as an optimization problem with 'ultra-high dimensionality-15000, prohibitive evaluation costs (40–70 minutes per test), and extreme system fragility' and propose semantic mismatch later, while formalizing the search space and objectives in Sec.2. The PerceptOS framework is formally formalized in Alg.1. The theoretical analysis is in App.D. These were recognized by Reviewers **beJG** and **xsCP**, who noted our well-defined problem, and Reviewer **G5oA**, who praised its clarity.
> > >
> > > **Novelty**:  PerceptOS is **the first to propose deep semantic perception to solve semantic mismatch** in existing LLM methods which are based on heuristic reasoning. A naive method—having  LLM compare numerical performance shifts and configuration edits across iterations—struggles with severe attribution errors in high-dimensional spaces , resulting in a shallow perception that  fails to capture actual systemic changes. The table show difference in perception from traditional methods:
> > >
> > > |Method|Semantic Perception|Issues in High-Dim/Risk/Cost Tasks|
> > > |:-|:-|:-|
> > > |CostModel|No (Numeric-only)|fails on sparse samples|
> > > |RL|No (Vector-only)|High interaction cost|
> > > |Monte Carlo|No (Numeric-only)|High interaction cost|
> > > |Bayesian Opt.|No (Numeric-only)|Fails at 15k+ dims; risk of boot-failure|
> > > |LLM (AutoOS/BYOS)|No (Numeric-only)|Severe Semantic Mismatch|
> > >
> > > PerceptOS is submitted in application track and offers reference takeaways in these aspects:
> > >
> > > Semantic perception paradigm can be effective in LLM for optimization
> > >
> > > Perception module demonstrates how to precisely and generally capture underlying workload semantics
> > >
> > > BHIT assists LLMs in aligning with high-dimensional configuration spaces, capable of modeling any configuration space with dependencies
> > >
> > > Posterior Enhancement proposes an method to have a chance to recover possible semantic mismatches
> > >
> > > Semantic Closed-Loop is first to capture semantic shifts in system states post-modification and optimize accordingly
> > >
> > > After clarification, reviewer **bEJG** acknowledge the "first closed-loop perception framework" . reviewer **G5oA** and **xsCP**  acknowledged the innovation that this work **'enabling a paradigm shift from “black-box search” to “perception-driven tuning"** in 'LLM for optimization'
> > >
> > > We believe the number of clarifying experiments is reasonable in rebuttal and do not affect our core claims. We hope it solves the concerns

---

### Decision · Program_Chairs · 2026-04-30

**Decision:**

Accept (regular)

**Comment:**

The paper introduces PerceptOS, a perception-driven framework for automated kernel configuration tuning targeting OS-intensive workloads. It aims to address the semantic mismatch between LLM reasoning and low-level system behavior through three modules: (1) a Perception Module that aligns telemetry into semantic fingerprints, (2) a Global Search Module utilizing a Bi-level Hierarchical Induction Tree, and (3) a Posterior Enhancement Module to suppress hallucinations. Experiments show PerceptOS demonstrates throughput gains and generally outperforms prior LLM-based baselines within a limited iteration budget.

Reviewers generally acknowledge that the semantic mismatch is the core bottleneck of existing methods, and the proposed method is practical and technically solid, making large-scale kernel search tractable. The empirical evaluation is also satisfactory,
spanning multiple workloads, hardware environments, OS distributions, and LLMs. Some concerns regarding ML novelty, baseline fairness, and robustness, and experimental details are raised. The authors have presented a rebuttal, which addresses most of the concerns. Overall, I think the paper makes a solid contribution towards kernel optimization, and the overall solution is satisfactory, despite that the components of the model are somewhat moderate. I recommend acceptance and encourage the authors to revise the paper based on the reviewer comments.